# DISTRIBUTED ONLINE OPTIMIZATION WITH LONG-TERM CONSTRAINTS

## ABSTRACT

We consider distributed online convex optimization problems, where the distributed system consists of various computing units connected through a time-varying communication graph. In each time step, each computing unit selects a constrained vector, experiences a loss equal to an arbitrary convex function evaluated at this vector, and may communicate to its neighbors in the graph. The objective is to minimize the system-wide loss accumulated over time. We propose a decentralized algorithm with regret and cumulative constraint violation in $\mathcal{O}(T^{\max\{c,1-c\}})$ and $\mathcal{O}(T^{1-c/2})$, respectively, for any $c \in (0,1)$, where $T$ is the time horizon. When the loss functions are strongly convex, we establish improved regret and constraint violation upper bounds in $\mathcal{O}(\log(T))$ and $\mathcal{O}(\sqrt{T \log(T)})$. These regret scalings match those obtained by state-of-the-art algorithms and fundamental limits in the corresponding centralized online optimization problem (for both convex and strongly convex loss functions). In the case of bandit feedback, the proposed algorithms achieve a regret and constraint violation in $\mathcal{O}(T^{\max\{c,1-c/3\}})$ and $\mathcal{O}(T^{1-c/2})$ for any $c \in (0,1)$. We numerically illustrate the performance of our algorithms for the particular case of distributed online regularized linear regression problems.

## 1 INTRODUCTION

The Online Convex Optimization (OCO) paradigm Hazan (2016) has recently become prominent in various areas of machine learning where the environment sequentially generating data is too complex to be efficiently modeled. OCO portrays optimization as a process, and applies a robust and sequential optimization approach where one learns from experiences as time evolves. Specifically, under the OCO framework, at each time-step the learner commits to a decision and suffers from a loss, a convex function of the decision. The successive loss functions are unknown beforehand and may vary arbitrarily over time. At the end of each step, the loss function may be revealed (a scenario referred to as *full information*). Alternatively, the experienced loss only might be available (*bandit feedback*). The objective of the decision maker is to minimize the loss accumulated over time. The performance of an algorithm in OCO is assessed through the notion of regret, comparing the accumulated loss under the algorithm and that achieved by an Oracle always selecting the best fixed decision. In case of full information feedback, it is known that the best possible regret scales in $\mathcal{O}(\sqrt{T})$ (resp. $\mathcal{O}(\log T)$) for convex (resp. strongly convex) loss functions Zinkevich (2003); Hazan et al. (2007); Abernethy et al. (2009).

This paper extends the OCO framework to a distributed setting where (different) data is collected and processed at $N$ computing units in a network. More precisely, we consider scenarios where in each time-step, each unit $i$ commits to a decision $\mathbf{x}_i(t)$ and then experiences a *local* loss equal to $\ell_{i,t}(\mathbf{x}_i(t))$. Units update their decision based on previously observed losses and messages received from neighboring units with the objective of identifying the decision $\mathbf{x}^\star = \arg\min_\mathbf{x} \sum_{t=1}^T \sum_{i=1}^N \ell_{i,t}(\mathbf{x})$ minimizing the accumulated system-wide loss. Many traditional applications of the centralized OCO framework Hazan (2016) naturally extend to this distributed setting. As a motivating example, consider the following distributed online spam filtering task (refer to Hazan (2016) for a description of the spam filter design problem in a centralized setting). In each time-step, each unit $i$ (here an email server) receives an email characterized by a vector $\mathbf{a}_{i,t} \in \mathbb{R}^d$ (according to the "bag-of-words" representation). Unit $i$ applies for this email a filter represented

by a vector $\mathbf{x}_i(t) \in \mathcal{X}$ where $\mathcal{X}$ is convex compact subset of $\mathbb{R}^d$, returns a label $f(\mathbf{a}_{i,t}^\top \mathbf{x}_i(t))$, and experiences a loss equal to $\ell_{i,t}(\mathbf{x}_i(t)) = (f(\mathbf{a}_{i,t}^\top \mathbf{x}_i(t)) - y_{i,t})^2$ where $y_{i,t}$ is the true email label (-1 for spam or 1 for valid). Note that the sequences of loss functions are inherently different at various units because the latter receive different emails. Nevertheless, each unit would ideally wish to identify and apply as fast as possible the filter minimizing the system-wide loss, i.e., a filter that exploits the knowledge extracted from *all* emails, including those received at other units. By leveraging this knowledge, each unit would adapt faster to an adversary also modifying in an online manner spam emails. More generally the distributed OCO framework can be applied to networks of learning agents, where each agent wishes to take advantage of what other agents have learnt to speed up and robustify its own learning process.

## 1.1 THE DOCO (DISTRIBUTED ONLINE CONVEX OPTIMIZATION) FRAMEWORK

We describe here our distributed optimization problem in more detail. We consider a network of $N$ computing units described by a sequence of directed graphs $\mathcal{G}_t = \{\mathcal{V}, \mathcal{E}_t\}$ with node set $\mathcal{V} = \{1, \ldots, N\}$ and edge set $\mathcal{E}_t$ at time $t$. $\mathcal{G}_t$ represents the communication constraints at the end of time-step $t$: each unit is allowed to send its decision at time $t$ to its neighbors in $\mathcal{G}_t$. Each unit $i \in \mathcal{V}$ is associated with a sequence of convex loss functions $\{\ell_{i,t}\}_{t=1}^T$, where $\ell_{i,t} : \mathbb{R}^d \to \mathbb{R}$.

**Optimization process.** In each time-step $t$, each unit $i \in \mathcal{V}$ selects $\mathbf{x}_i(t) \in \mathbb{R}^d$. Then, in case of full information feedback, the loss function $\ell_{i,t}$ is revealed to unit $i$, whereas in case of bandit feedback, the loss $\ell_{i,t}(\mathbf{x}_i(t))$ is revealed only. Unit $i$ finally receives vectors, functions of decisions selected by its neighbors in $\mathcal{G}_t$, i.e., $\mathbf{x}_j(t)$ for $j$ such that $(j, i) \in \mathcal{E}_t$, and updates its decision for the next time-step.

**Decision constraints.** The decisions should be selected in $\mathcal{X}$ a convex subset of $\mathbb{R}^d$ characterized by a family of inequalities: $\mathcal{X} = \{\mathbf{x} \in \mathbb{R}^d \mid c_s(\mathbf{x}) \leq 0, \ s = 1, \ldots, p\}$. Imposing such constrained decisions implies that each unit should be able in each time-step to perform a projection onto $\mathcal{X}$, which can be extremely computationally expensive. To circumvent this difficulty, we adopt the notion of *long-term constraints* introduced in Mahdavi et al. (2012). Specifically, we only impose that the constraints are satisfied in a long run rather than in each time-step, i.e., that $\sum_{t=1}^T \sum_{i=1}^N \sum_{s=1}^p c_s(\mathbf{x}_i(t)) \leq 0$. This relaxation allows units to violate the constraints by projecting onto a simpler set that contains $\mathcal{X}$. Our results can be modified to account for the actual constraints (but using projection steps).

**Regrets and cumulative absolute constraint violation.** The objective is to design distributed sequential decision selection algorithms so that each unit identifies the decision minimizing the accumulated system-wide loss. The performance of such an algorithm is hence captured by the regrets at the various units. The regret at unit $i$ is:

$$\mathsf{Reg}(i, T) := \sum_{t=1}^T \sum_{j=1}^N \ell_{j,t}(\mathbf{x}_i(t)) - \sum_{t=1}^T \sum_{j=1}^N \ell_{j,t}(\mathbf{x}^\star), \tag{1}$$

where $\mathbf{x}^\star = \arg\min_{\mathbf{x} \in \mathcal{X}} \sum_{t=1}^T \sum_{j=1}^N \ell_{j,t}(\mathbf{x})$. The system-level regret is defined as the worst possible regret at all units: $\mathsf{SReg}(T) \triangleq \max_{i=1,\ldots,N} \mathsf{Reg}(i, T)$. Now since we allow units to select decisions outside $\mathcal{X}$, the performance of an algorithm is further characterized by the so-called cumulative absolute constraint violation defined by: (here $[a]_+ = \max\{0, a\}$)

$$\mathsf{CACV}(T) := \sum_{t=1}^T \sum_{i=1}^N \sum_{s=1}^p [c_s(\mathbf{x}_i(t))]_+. \tag{2}$$

## 1.2 MAIN RESULTS

We propose simple distributed algorithms where in each time-step, each unit combines information received from its neighbors to update its decision and its local dual variable. Our algorithms enjoy the following performance guarantees:

**Full Information feedback.** In the case of full information feedback, the proposed algorithms achieve a system-level regret and a cumulative constraint violation in $\mathcal{O}(T^{\max\{1-c,c\}})$ and

$\mathcal{O}(T^{1-c/2})$, respectively and for any $c \in (0,1)$ ($c$ expresses the trade-off between regret and cumulative constraint violation). Theses bounds match those of centralized online optimization algorithms in Mahdavi et al. (2012); Jenatton et al. (2016); Yuan & Lamperski (2018). When $c = 1/2$, we get a regret scaling in $\mathcal{O}(\sqrt{T})$, which corresponds to the fundamental regret limits for centralized online problems Abernethy et al. (2009), which is rather surprising in view of the dynamically changing environment, the decentralized structure of the algorithm, and the presence of the constraints. When the loss functions are strongly convex, we establish improved upper bounds on the regret and cumulative connstraint violation in $\mathcal{O}(\log(T))$ and $\mathcal{O}(\sqrt{T \log(T)})$. These bounds generalize to our distributed setting those derived in Yuan & Lamperski (2018) for centralized problems.

**Bandit feedback.** In the case of bandit feedback, the proposed algorithms achieve a system-level regret and a cumulative constraint violation in $\mathcal{O}(d^2 T^{\max\{1-c/3,c\}})$ and $\mathcal{O}(dT^{1-c/2})$, respectively, for any $c \in (0,1)$. For example, when $c = \frac{3}{4}$, the proposed algorithm attains a regret bound in $\mathcal{O}(d^2 T^{3/4})$. The performance guarantees can be improved to $\mathcal{O}(d^2 T^{2/3} \log(T))$ and $\mathcal{O}(d\sqrt{T \log(T)})$ in the case of strongly convex losses.

## 1.3 RELATED WORK

Early work on online convex optimization in a centralized setting include Zinkevich (2003); Flaxman et al. (2005). Today we know that a regret in $\mathcal{O}(\sqrt{T})$ is achievable in both full information and bandit feedback, see e.g. Bubeck et al. (2017). Projection-free algorithms have been also developed Mahdavi et al. (2012); Jenatton et al. (2016); Yuan & Lamperski (2018) with regret and cumulative constraint violation in $\mathcal{O}(T^{\max\{c,1-c\}})$ and $\mathcal{O}(T^{1-c/2})$ ($c \in (0,1)$) in case of full information feedback (Yuan & Lamperski (2018) uses the cumulative squared constraint violation). Our algorithms achieve the same guarantees in a distributed setting.

It is worth zooming into the rich literature on centralized online convex optimization with bandit feedback. In the seminal work Flaxman et al. (2005), the authors designed an algorithm with one-point bandit feedback and regret in $\mathcal{O}(d^2 T^{3/4})$. The work Agarwal et al. (2010) extended this algorithm to multi-point bandit feedback setting, where multiple points around the decision can be queried for the loss function; they established $\mathcal{O}(d^2 \sqrt{T})$ and $\mathcal{O}(d^2 \log(T))$ regret bounds for general convex and strongly convex loss functions, respectively. The work Mahdavi et al. (2012) studied the online bandit optimization with long-term constraints under two-point bandit feedback for domain. They established $\mathcal{O}(\sqrt{T})$ and $\mathcal{O}(d^2 T^{3/4})$ bounds on the regret and the cumulative constraint violations, respectively. In this paper, we design distributed algorithms with one-point bandit feedback only, and with the same regret guarantees as the centralized algorithm in Flaxman et al. (2005).

Over the last few years, there have been a rising interest for the Distributed OCO framework. Particularly, Shahrampour & Jadbabaie (2017); Lee et al. (2017) propose distributed algorithms with $\mathcal{O}(\sqrt{T})$ regret, but require an exact projection onto the decision set in each time-step. Zhang et al. (2017) presents a distributed online conditional gradient algorithm, replacing the projection steps by a much simpler linear optimization steps, but at the expense of worse and sub-optimal regret guarantees, scaling in $\mathcal{O}(T^{3/4})$. The other approach to avoid projections is to allow the algorithm to violate the constraints, and has been studied in Yuan et al. (2018). The problem studied in Yuan et al. (2018) is a special case of our problem (where only one inequality constraint is considered), and the regret and cumulative constraint violation guarantees obtained there are much worse than ours. The authors of Li et al. (2018); Yi et al. (2019) also use the long-term constraints approach to avoid projections, but analyze a very different optimization problem where units have different decision variables, and no consensus among units is required. Finally it is worth mentioning that all the aforementioned papers are restricted to full information feedback.

**Notation and Terminology.** Let $\|\mathbf{x}\|$ and $[\mathbf{x}]_i$ to denote the Euclidean norm and the $i$th component of a vector $\mathbf{x} \in \mathbb{R}^d$, respectively. Let $\Pi_{\mathcal{X}}[\mathbf{x}]$ be the Euclidean projection of a vector $\mathbf{x}$ onto the set $\mathcal{X}$. Let $\mathbb{R}_+^p$ be the nonnegative orthant in $\mathbb{R}^p$: $\mathbb{R}_+^p = \{\mathbf{x} \in \mathbb{R}^p \mid [\mathbf{x}]_i \geq 0, i = 1, \ldots, p\}$. Denote the $(i,j)$-th element of a matrix $\mathbf{A}$ by $[\mathbf{A}]_{ij}$. For a convex function $f$, a subgradient (resp. gradient when $f$ is differentiable) at a point $\mathbf{x}$ is denoted by $\partial f(\mathbf{x})$ (resp. $\nabla f(\mathbf{x})$). Given two positive sequences $\{a_t\}_{t=1}^\infty$ and $\{b_t\}_{t=1}^\infty$, we write $a_t = \mathcal{O}(b_t)$ if $\limsup_{t \to \infty} a_t/b_t < \infty$.

## 2 FULL-INFORMATION FEEDBACK

In this section, we focus on the case of full-information feedback, where at the end of each time-step, the entire loss function $\ell_{i,t}$ is revealed to unit $i$. More precisely, unit $i$ has access to the gradient of the loss function $\ell_{i,t}$ at any query point. We make the following assumptions, which are standard in the literature e.g., Mahdavi et al. (2012); Jenatton et al. (2016); Yuan & Lamperski (2018); Nedic & Ozdaglar (2009); Yan et al. (2013); Duchi et al. (2012); Hosseini et al. (2013); Nedic et al. (2010).

**Assumption 1** $\mathcal{X} \subseteq \mathcal{B} := \left\{ \mathbf{x} \in \mathbb{R}^d \mid \|\mathbf{x}\| \leq R_{\mathcal{X}} \right\}$ *with* $R_{\mathcal{X}} > 0$.

**Assumption 2** *The functions* $\ell_{i,t}$ *and* $c_s$ *are convex with bounded gradients:*
$$\max_{i=1,\ldots,N} \max_{t=1,\ldots,T} \max_{\mathbf{x} \in \mathcal{B}} \|\nabla \ell_{i,t}(\mathbf{x})\| \leq G_\ell, \quad \max_{s=1,\ldots,p} \max_{\mathbf{x} \in \mathcal{B}} \|\nabla c_s(\mathbf{x})\| \leq G_c.$$
*We let* $G = \max\{G_\ell, G_c\}$.

**Assumption 3** *There exists an integer* $B \geq 1$ *such that the union graph* $(\mathcal{V}, \mathcal{E}_{kB+1} \cup \cdots \cup \mathcal{E}_{(k+1)B})$ *is strongly connected for all* $k \geq 0$.

**Assumption 4** *Associated with* $\mathcal{G}_t$ *there is the weight matrix* $\mathbf{A}(t)$ *which satisfies for all* $t \geq 1$:
*(i)* $\mathbf{A}(t)$ *is doubly stochastic for all* $t \geq 1$, *i.e.,* $\sum_{j=1}^{N} [\mathbf{A}(t)]_{ij} = 1$ *and* $\sum_{i=1}^{N} [\mathbf{A}(t)]_{ij} = 1, \forall i, j \in \mathcal{V}$;
*(ii) There exists a scalar* $\zeta > 0$ *such that* $[\mathbf{A}(t)]_{ii} \geq \zeta$ *for all* $i$ *and* $t \geq 1$, *and* $[\mathbf{A}(t)]_{ij} \geq \zeta$ *if* $(j, i) \in \mathcal{E}_t$ *and* $[\mathbf{A}(t)]_{ij} = 0$ *for all* $j$ *otherwise.*

Assumption 4 is quite standard in the literature on distributed online or offline optimization, and easy to achieve in a distributed manner in real-world networks. For example, when bidirectional communication between nodes is allowed, we can enforce symmetry on the node interaction matrix, which immediately makes it doubly stochastic. There are also other methods to construct doubly stochastic matrices for a network, see, e.g., Garin & Schenato (2010); Gharesifard & Cortés (2010).

---

**Algorithm 1** DOCO-LTC with full-information feedback

---

**Input:** Step sizes $\{\beta_t\}_{t=1}^T$, regularization parameters $\{\eta_t\}_{t=1}^T$
**Initialize:** $\mathbf{x}_i(1) = \mathbf{0} \in \mathbb{R}^d, \boldsymbol{\lambda}_i(1) = \mathbf{0} \in \mathbb{R}^p, \forall i = 1, \ldots, N$
1: **for** $t = 1$ to $T$ **do**
2:     Unit $i$ commits to a decision $\mathbf{x}_i(t)$, and then after receiving $\ell_{i,t}$, compute

$$\mathbf{y}_i(t) = \mathbf{x}_i(t) - \beta_t \left[ \nabla \ell_{i,t}(\mathbf{x}_i(t)) + \sum_{s=1}^{p} [\boldsymbol{\lambda}_i(t)]_s \partial [c_s(\mathbf{x}_i(t))]_+ \right]$$

3:     Unit $i$ communicates $\mathbf{y}_i(t)$ to its neighbors and updates its decision as

$$\mathbf{x}_i(t+1) = \Pi_{\mathcal{B}}\left(\mathbf{p}_i(t)\right), \quad \text{where} \quad \mathbf{p}_i(t) = \sum_{j=1}^{N} [\mathbf{A}(t)]_{ij} \mathbf{y}_j(t)$$

4:     Unit $i$ updates its dual variable: $\boldsymbol{\lambda}_i(t+1) = \arg\max_{\boldsymbol{\lambda} \in \mathbb{R}_+^d} \mathsf{L}_{i,t}((\mathbf{x}_i(t+1), \boldsymbol{\lambda})$
5: **end for**

---

The pseudo-code of our algorithm, DOCO-LTC (LTC stands for Long-Term Constraints), is presented in Algorithm 1. It generalizes the algorithm in Yuan & Lamperski (2018) to our distributed setting. In contrast to the literature on (online) distributed optimization with inequality constraints Yuan et al. (2018); Li et al. (2018); Khuzani & Li (2016), the algorithm does not need to maintain an iterative dual update process for every unit, which can be computed locally and explicitly. Moreover, no consensus updates on the dual variables are necessary, reducing the communication complexity.

The design and convergence analysis of DOCO-LTC rely on the following *online augmented Lagrangian function* associated with unit $i \in \mathcal{V}$: for $t \geq 1$,

$$\mathsf{L}_{i,t}(\mathbf{x}, \boldsymbol{\lambda}) \triangleq \ell_{i,t}(\mathbf{x}) + \sum_{s=1}^{p} [\boldsymbol{\lambda}]_s [c_s(\mathbf{x})]_+ - \frac{\eta_t}{2} \|\boldsymbol{\lambda}\|^2, \tag{3}$$

where $\boldsymbol{\lambda} = [[\boldsymbol{\lambda}]_1, \ldots, [\boldsymbol{\lambda}]_p]^\mathsf{T} \in \mathbb{R}^p_+$ is the vector of Lagrangian multipliers with $[\boldsymbol{\lambda}]_s$ being associated with the $s$th inequality constraint $c_s(\mathbf{x}) \le 0$ and $\eta_t$ is the regularization parameter. We note that:

$$\nabla_\mathbf{x} \mathsf{L}_{i,t}(\mathbf{x}_i(t), \boldsymbol{\lambda}_i(t)) = \nabla \ell_{i,t}(\mathbf{x}_i(t)) + \sum_{s=1}^p [\boldsymbol{\lambda}_i(t)]_s \partial [c_s(\mathbf{x}_i(t))]_+,$$

where $\partial [c_s(\mathbf{x}_i(t))]_+$ can be calculated as follows for $s = 1, \ldots, p$:

$$\partial [c_s(\mathbf{x}_i(t))]_+ \quad = \quad \begin{cases} \nabla c_s(\mathbf{x}_i(t)), & \text{if } c_s(\mathbf{x}_i(t)) > 0 \\ 0, & \text{otherwise.} \end{cases}$$

Moreover, the dual update $\boldsymbol{\lambda}_i(t+1)$ in DOCO-LTC can be calculated explicitly as follows:

$$[\boldsymbol{\lambda}_i(t+1)]_s = \frac{[c_s(\mathbf{x}_i(t+1))]_+}{\eta_t}, \qquad\qquad s = 1, \ldots, p. \tag{4}$$

**Theorem 1 (Convex loss functions and full-information feedback)** *Under Assumptions 1–4, the regret and cumulative constraint violation of DOCO-LTC with parameters $\eta_t = \frac{1}{T^c}$ and $\beta_t = \frac{1}{apG^2 T^c}$ for some $c \in (0, 1)$, $a > 1$, and all $t \ge 1$, satisfy: for all $T \ge 1$,*

$$\mathsf{SReg}(T) \le \tilde{C} T^{\max\{1-c,c\}} \text{ and } \mathsf{CACV}(T) \le \bar{C} T^{1-c/2},$$

*where $\tilde{C} = \frac{1}{2} apN G^2 R_\mathcal{X}^2 + \frac{1}{ap} N(1 + \hat{C}) + \frac{N\hat{C}^2}{4a(a-1)p}$ with $\hat{C} = 2N \left( \frac{3N}{\psi^{2+1/B}(1-\psi^{1/B})} + 4 \right)$ and $\psi = \left( 1 - \frac{\zeta}{4N^2} \right)^{-2}$, and $\bar{C} = \sqrt{\frac{N^2}{a-1} \left( 1 + 2apGR_\mathcal{X} + \frac{1}{2} a^2 p^2 G^2 R_\mathcal{X}^2 \right)}$.*

Theorem 1 shows that DOCO-LTC has the same guarantees as those of the centralized algorithms in Mahdavi et al. (2012); Jenatton et al. (2016); Yuan & Lamperski (2018). The user-defined parameter $c$ tunes the trade-off between $\mathsf{SReg}$ and $\mathsf{CACV}$ (for $c = 1/2$, we get a regret and constraint violation in $\mathcal{O}(\sqrt{T})$ and $\mathcal{O}(T^{3/4})$).

**Communication cost vs. regret.** The communication cost, i.e., the number of vectors transmitted per round in the network, is simply equal to the number of edges in the network. Taking the case of $B = 1$ (i.e., the graph is fixed and connected) as an example, we can establish that the regret bound in Theorem 1 scales as $\mathcal{O}\left( \frac{N^4}{(1-\sigma_2(\mathbf{A}))^2} T^{\max\{c,1-c\}} \right)$, where $\sigma_2(\mathbf{A})$ is the second largest singular value of the weight matrix $\mathbf{A}$. If we choose the weight matrix as the maximum-degree weights (see, e.g., Yuan et al. (2019)), we have the following conclusions: i) *Random geometric graph:* the regret bound scales as $\frac{N^6}{\log^2(N)} T^{\max\{c,1-c\}}$ and at most $2 \log^{1+\epsilon}(N) N$ vectors are transmitted per round; ii) *$k$-regular expander graph:* $\sigma_2(\mathbf{A})$ is constant, the regret bounds scales as $N^4 T^{\max\{c,1-c\}}$ and $2kN$ vectors are transmitted per round; and iii) *complete graph:* $\sigma_2(\mathbf{A}) = 0$ and $N(N-1)$ vectors are transmitted per round.

Next we improve DOCO-LTC performance guarantees when the loss functions are strongly convex.

**Assumption 5** *The loss function $\ell_{i,t}$ is $\sigma$-strongly convex over $\mathcal{B}$, that is, for any $\mathbf{x}, \mathbf{y} \in \mathcal{B}$,*

$$\ell_{i,t}(\mathbf{x}) \ge \ell_{i,t}(\mathbf{y}) + \nabla \ell_{i,t}(\mathbf{y})^\mathsf{T}(\mathbf{x} - \mathbf{y}) + \frac{\sigma}{2} \|\mathbf{x} - \mathbf{y}\|^2.$$

**Theorem 2 (Strongly convex loss functions and full-information feedback)** *Under Assumptions 1–5, the regret and cumulative constraint violation of DOCO-LTC with parameters $\eta_t = \frac{2pG^2}{\sigma t}$ and $\beta_t = \frac{1}{\sigma t}$ for all $t \ge 1$, satisfy: for all $T \ge 3$,*

$$\mathsf{SReg}(T) \le \tilde{C}_\text{sc} \log(T), \quad \text{and} \quad \mathsf{CACV}(T) \le \bar{C}_\text{sc} \sqrt{T \log(T)},$$

*where $\tilde{C}_\text{sc} = \frac{NG^2}{2\sigma}(4 + 4\hat{C} + \hat{C}^2)$ ($\hat{C}$ is shown in Theorem 1) and $\bar{C}_\text{sc} = \frac{4pNG^{3/2}}{\sqrt{\sigma}} \left( \sqrt{R_\mathcal{X}} + \sqrt{\frac{G}{\sigma}} \right)$.*

In the case of strongly convex loss functions, the regret and constraint violation guarantees of DOCO-LTC also match those obtained by the centralized algorithm in Yuan & Lamperski (2018). Note that one cannot actually get a better regret scaling, even in the centralized setting Abernethy et al. (2009).

## 3   ONE-POINT BANDIT FEEDBACK

This section is devoted to the case of bandit feedback, where at the end of each time-step, unit $i$ can observe the value of the loss function $\ell_{i,t}$ at only one point around $\mathbf{x}_i(t)$. The pseudo-code of our algorithm adapted to this feedback is presented in Algorithm 2.

---

**Algorithm 2** DOCO-LTC with one-point bandit feedback

---

**Input:** Step sizes $\{\beta_t\}_{t=1}^T$, regularization parameters $\{\eta_t\}_{t=1}^T$, exploration parameters $\{\varepsilon_t\}_{t=1}^T$, and shrinkage parameter $\pi$

**Initialize:** $\mathbf{x}_i(1) = \mathbf{0} \in \mathbb{R}^d, \boldsymbol{\lambda}_i(1) = \mathbf{0} \in \mathbb{R}^p, \forall i = 1, \ldots, N$

1: **for** $t = 1$ to $T$ **do**
2:     Unit $i$ commits to a decision $\mathbf{x}_i(t)$, and then observes the loss $\ell_{i,t}(\mathbf{x}_i(t) + \varepsilon_t \mathbf{u}_i(t))$ where $\mathbf{u}_i(t)$ is randomly chosen on the unit sphere ($\|\mathbf{u}_i(t)\| = 1$)
3:     Unit $i$ builds the following **one-point gradient estimator**:

$$\tilde{\nabla}\ell_{i,t}(\mathbf{x}_i(t)) = \frac{d}{\varepsilon_t}\ell_{i,t}(\mathbf{x}_i(t) + \varepsilon_t \mathbf{u}_i(t))\mathbf{u}_i(t)$$

and computes

$$\mathbf{y}_i(t) = \mathbf{x}_i(t) - \beta_t \left[\tilde{\nabla}\ell_{i,t}(\mathbf{x}_i(t)) + \sum_{s=1}^p [\boldsymbol{\lambda}_i(t)]_s \partial[c_s(\mathbf{x}_i(t))]_+\right]$$

4:     Unit $i$ updates its decision using $\mathbf{y}_j(t)$ received from its neighbors as

$$\mathbf{x}_i(t+1) = \Pi_{\mathcal{B}}(\mathbf{p}_i(t)), \quad \text{where} \quad \mathbf{p}_i(t) = \sum_{j=1}^N [\mathbf{A}(t)]_{ij}\mathbf{y}_j(t)$$

5:     Node $i$ updates its dual variable $\boldsymbol{\lambda}_i(t+1) = \arg\max_{\boldsymbol{\lambda} \in \mathbb{R}_+^d} \tilde{\mathsf{L}}_{i,t}((\mathbf{x}_i(t+1), \boldsymbol{\lambda})$
6: **end for**

---

The design and convergence analysis of our algorithm here rely on the *smoothed* version $\tilde{\mathsf{L}}_{i,t}(\mathbf{x}, \boldsymbol{\lambda})$ of the online augmented Lagrangian function (3), i.e., $\tilde{\mathsf{L}}_{i,t}(\mathbf{x}, \boldsymbol{\lambda})$: for $t \geq 1$,

$$\tilde{\mathsf{L}}_{i,t}(\mathbf{x}, \boldsymbol{\lambda}) \triangleq \tilde{\ell}_{i,t}(\mathbf{x}; \varepsilon) + \sum_{s=1}^p [\boldsymbol{\lambda}]_s[c_s(\mathbf{x})]_+ - \frac{\eta_t}{2}\|\boldsymbol{\lambda}\|^2, \tag{5}$$

where $\tilde{\ell}_{i,t}(\mathbf{x}; \varepsilon) = \mathbb{E}_{\mathbf{v}}[\ell_{i,t}(\mathbf{x} + \varepsilon\mathbf{v})]$ is the smoothed loss function, and $\mathbf{v}$ is a vector uniformly distributed over the unit sphere. As in the case of full information feedback, the dual update $\boldsymbol{\lambda}_i(t+1)$ can be calculated explicitly according to (4).

In the case of bandit feedback, we need to introduce the shrinkage parameter $\pi$ to ensure that the random query point $\mathbf{x}_i(t) + \varepsilon_t \mathbf{u}_i(t)$ belongs to the set $\mathcal{B}$. Indeed, we have:

$$\|\mathbf{x}_i(t) + \varepsilon_t \mathbf{u}_i(t)\| \leq \|\mathbf{x}_i(t)\| + \varepsilon_t\|\mathbf{u}_i(t)\| \leq (1 - \pi)R_{\mathcal{X}} + \varepsilon_t \leq R_{\mathcal{X}}$$

where the second inequality follows from the fact that $\mathbf{x}_i(t) \in (1 - \pi)\mathcal{B}$ and $\|\mathbf{u}_i(t)\| = 1$ and the last inequality holds when $\varepsilon_t \leq \pi R_{\mathcal{X}}$.

To establish upper bounds on the regret and cumulative constraint violation of our algorithm, we make the following standard assumption on the loss functions $\ell_{i,t}(\mathbf{x})$ (commonly adopted even in centralized online bandit optimization Flaxman et al. (2005)).

**Assumption 6** *The loss functions $\ell_{i,t}(\mathbf{x})$ are uniformly bounded over $\mathcal{B}$:*

$$\sup_{\mathbf{x} \in \mathcal{B}} \max_{i=1,\ldots,N} \max_{t=1,\ldots,T} |\ell_{i,t}(\mathbf{x})| \leq C.$$

Since algorithms for bandit feedback are inherently randomized, we investigate averaged versions of the regret and the cumulative constraint violation: $\mathsf{E\text{-}SReg}(T) := \max_{i=1,\ldots,N} \mathbb{E}[\mathsf{Reg}(i,T)]$ and $\mathsf{E\text{-}CACV}(T) := \sum_{t=1}^T \sum_{i=1}^N \sum_{s=1}^p \mathbb{E}[[c_s(\mathbf{x}_i(t))]_+]$.

**Theorem 3 (Convex functions with bandit feedback)** *Under Assumptions 1–4 and 6, the regret and cumulative constraint violation of DOCO-LTC with parameters*

$$\eta_t = \frac{1}{T^c}, \quad \beta_t = \frac{1}{apG^2T^c}, \quad \varepsilon_t = \frac{1}{T^b}, \quad \pi = \frac{1}{R_{\mathcal{X}}T^b}$$

*for some $c \in (0, 1)$, $b = c/3$ and all $t \geq 1$, satisfy: for all $T \geq 1$,*

$$\mathsf{E\text{-}SReg}(T) \leq \tilde{C}^{\S}T^{\max\{1-c/3,c\}} \quad \text{and} \quad \mathsf{E\text{-}CACV}(T) \leq \bar{C}^{\S}T^{1-c/2},$$

*where $\tilde{C}^{\S} = 3NG + \frac{NC\hat{C}d}{apG} + \frac{NC^2d^2}{apG^2} + \frac{1}{2}apNG^2R_{\mathcal{X}}^2 + \frac{N\hat{C}^2}{4a(a-1)p}$ ($\hat{C}$ is shown in Theorem 1) and $\bar{C}^{\S} = \sqrt{\frac{N^2}{a-1}\left(\frac{C^2d^2}{G^2} + 2apGR_{\mathcal{X}} + \frac{1}{2}a^2p^2G^2R_{\mathcal{X}}^2\right)}$.*

Note that DOCO-LTC achieves a regret scaling as $T^{3/4}$ when $c = \frac{3}{4}$, which is identical to that of centralized online bandit optimization Flaxman et al. (2005). This is rather remarkable considering the decentralized nature of the algorithm. Again, we can improve our bounds in the case of strongly convex loss functions.

**Theorem 4 (Strongly convex functions with bandit feedback)** *Under Assumptions 1–6, the regret and cumulative constraint violation of DOCO-LTC with parameters*

$$\eta_t = \frac{2pG^2}{\sigma t}, \quad \beta_t = \frac{1}{\sigma t}, \quad \varepsilon_t = \frac{1}{T^b}, \quad \pi = \frac{1}{R_{\mathcal{X}}T^b}$$

*for $b = \frac{1}{3}$, and all $t \geq 1$, satisfy: for all $T \geq 3$,*

$$\mathsf{E\text{-}SReg}(T) \leq \tilde{C}_{\mathrm{sc}}^{\S}T^{2/3}\log(T) \quad \text{and} \quad \mathsf{E\text{-}CACV}(T) \leq \bar{C}_{\mathrm{sc}}^{\S}\sqrt{T\log(T)},$$

*where $\tilde{C}_{\mathrm{sc}}^{\S} = 3NG + \frac{N}{2\sigma}\left(4C\hat{C}Gd + 4C^2d^2 + \hat{C}^2G^2\right)$ ($\hat{C}$ is shown in Theorem 1) and $\bar{C}_{\mathrm{sc}}^{\S} = \frac{4pNG}{\sqrt{\sigma}}\left(\sqrt{GR_{\mathcal{X}}} + \frac{Cd}{\sqrt{\sigma}}\right)$.*

## 4 NUMERICAL EXPERIMENT

We illustrate the performance of the proposed algorithms using a simple experiment. Specifically, we consider distributed online regularized linear regression problem over a network, formulated as follows:

$$\begin{array}{ll} \text{minimize} & \sum_{t=1}^{T}\sum_{i=1}^{N}\frac{1}{2}\left(\mathbf{a}_i(t)^{\mathsf{T}}\mathbf{x} - b_i(t)\right)^2 + \rho\|\mathbf{x}\|^2 \\ \text{subject to} & c_m(\mathbf{x}) = L - [\mathbf{x}]_m \leq 0, \quad m = 1, \ldots, d \\ & c_{d+m}(\mathbf{x}) = [\mathbf{x}]_m - U \leq 0, \quad m = 1, \ldots, d \end{array} \quad (6)$$

where $\rho \geq 0$ denotes the regularization parameter. The data $(\mathbf{a}_i(t), b_i(t)) \in \mathbb{R}^d \times \mathbb{R}$ is revealed only to unit $i$ at time $t$.

**Results on Synthetic Data.** Every entry of $\mathbf{a}_i(t)$ is generated uniformly at random within the interval $[-1, 1]$ and $b_i(t)$ is generated according to

$$b_i(t) = \mathbf{a}_i(t)^{\mathsf{T}}\bar{\mathbf{x}} + \epsilon_i(t)$$

where $[\bar{\mathbf{x}}]_i = 1$, for all $1 \leq i \leq \lfloor d/2 \rfloor$ and 0 otherwise, and the noise $\epsilon_i(t) \sim \mathcal{N}(0, 1)$. Throughout the experiments, we implement our algorithms over a time-varying directed network depicted in Fig. 1: the network is not connected in every time-step, but the union graph of any two consecutive time instances is strongly connected, that is, we have $B = 2$ in Assumption 3. The weight matrices associated with the networks in Fig. 1 are generated according to the maximum-degree weights (see, e.g., Yuan et al. (2019)). We set the parameters as follows: $N = 6$, $d = 4$, $L = -0.15$, $U = 0.15$, and $R_{\mathcal{X}} = U\sqrt{d}$. The performance of DOCO-LTC is averaged over 10 runs.

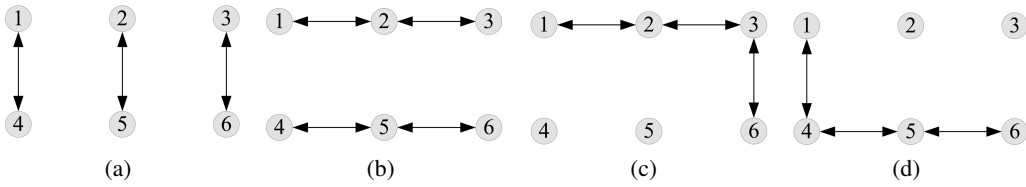

Figure 1: The network switches sequentially in a round robin manner between (a), (b), (c), and (d).

To get (not strongly) convex loss functions, we set $\rho = 0$. We run Algorithm 1 and Algorithm 2 with $c = 1/2$ and $c = 3/4$ and plot the maximum regret $\max_{i \in \mathcal{V}} \mathsf{Reg}(i, T)$, and $\mathsf{CACV}(T)$ as a function of the time horizon $T$ in Fig. 2(a) and Fig. 2(b), respectively. It can be seen from Fig. 2(a) that in the case of full-information feedback, the regret is smaller for $c = 1/2$, while in bandit feedback setting, the regret is smaller for $c = 3/4$. This is because $c = 1/2$ and $c = 3/4$ correspond to a *balanced* regret in the full-information setting and bandit feedback setting. By balanced, we mean that $1 - c = c$ in $T^{\max\{1-c,c\}}$ in Theorem 1 and $1 - c/3 = c$ in $T^{\max\{1-c/3,c\}}$ in Theorem 3, respectively. From Fig. 2(b) we also observe that for both feedback models, $\mathsf{CACV}$ is smaller for a larger value of $c$, i.e., $c = 3/4$. This is in compliance with the results established in Theorems 1 and 3. Finally, the performance is really degraded when going from full information to bandit feedback. This was also expected.

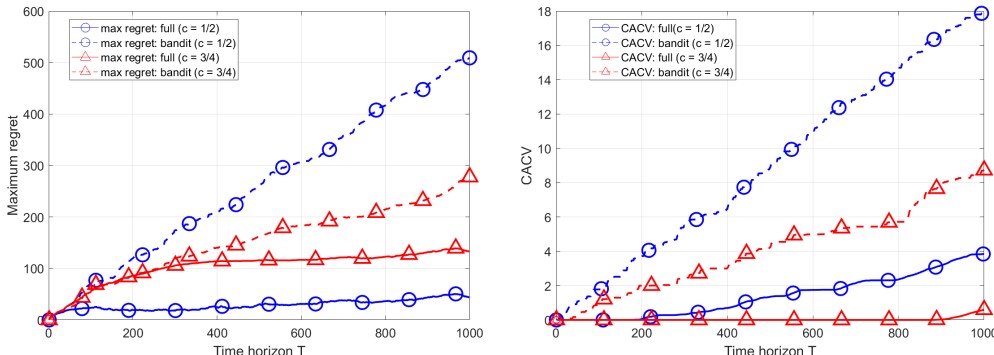

Figure 2: $\mathsf{SReg}$ and $\mathsf{CACV}$ vs. time for convex costs.

In the case of strongly convex losses, we run Algorithm 1 and Algorithm 2 with $\rho > 0$, namely $\rho = 1$ and $\rho = 2$. We plot the performance of the algorithms as a function the time horizon in Fig. 3(a) and Fig. 3(b), respectively. From Fig. 3, we confirm that the cost of bandit feedback is rather high. We also observe the regret and the violation constraint are smaller and flatter than those achieved of non-strongly convex loss functions ($\rho = 0$), for both feedback models. All these observations comply with the results established in Theorems 2 and 4.

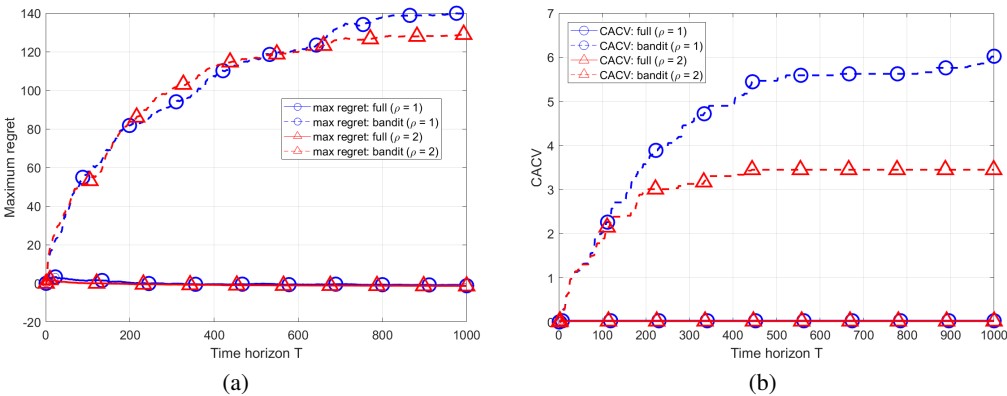

Figure 3: $\mathsf{SReg}$ and $\mathsf{CACV}$ vs. time for strongly convex costs.

**Results on Real Datasets.** We demonstrate the efficiency of our proposed algorithms on two real datasets selected from the LIBSVM[1] repository. The details of the datasets are summarized in Table 1.

We use the same network and parameters as those used in the synthetic data and let $c = 1/2$. For each dataset, we run Algorithm 1 and Algorithm 2 with $\rho = 0$ and $\rho = 1$, respectively. We plot

---

[1] https://www.csie.ntu.edu.tw/~cjlin/libsvmtools/datasets/

Table 1: Summary of datasets

| dataset | # of features | # of instances |
|---------|---------------|----------------|
| mg | 6 | 1385 |
| bodyfat | 14 | 252 |

the performance of the algorithms as a function the time horizon in Fig. 4 (mg dataset) and Fig. 5 (bodyfat dataset), respectively. These numerical experiments on real-world datasets show the convergence of the proposed algorithms and are consistent with the results established in Theorems 1–4. Finally, the performance is really degraded when going from strongly convex loss functions to convex loss functions.

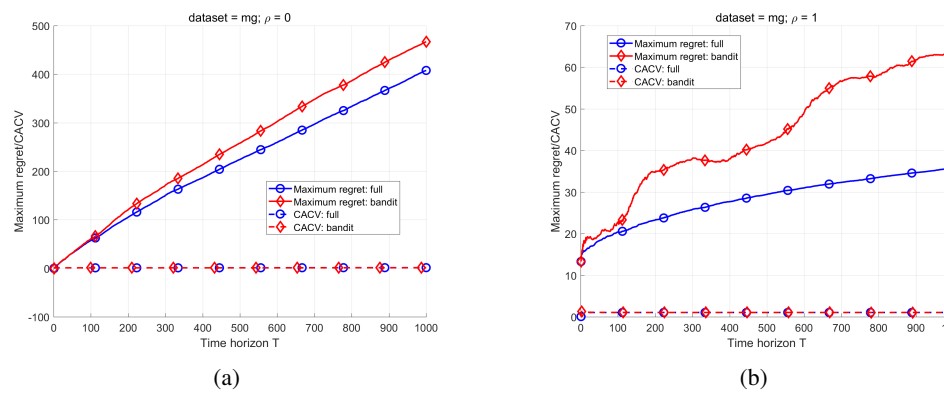

Figure 4: SReg and CACV vs. time for mg dataset.

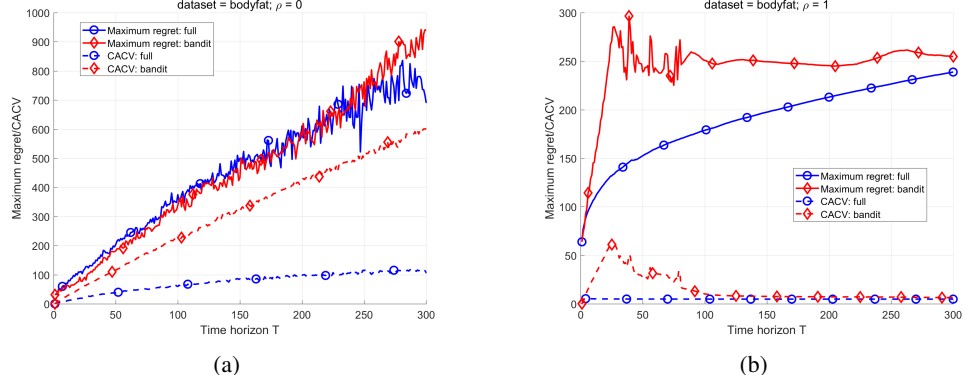

Figure 5: SReg and CACV vs. time for bodyfat dataset.

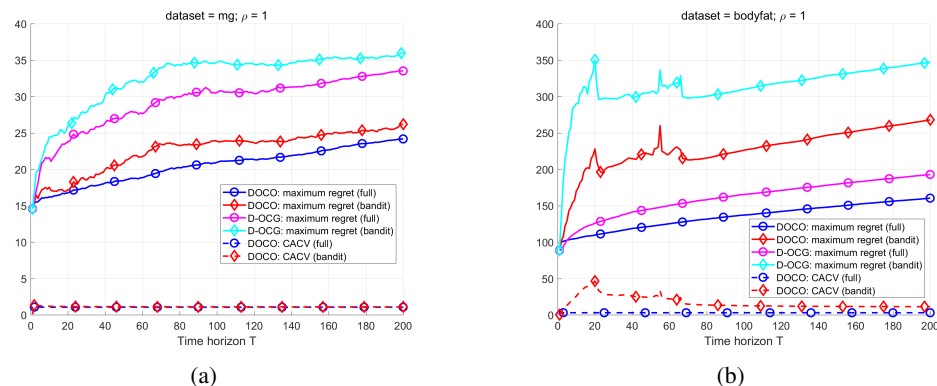

Figure 6: Comparison of the proposed algorithms with D-OCG on mg and bodyfat datasets.

We finally make comparisons with a standard distributed online projection-free algorithm (D-OCG in Zhang et al. (2017)) using mg and bodyfat real datasets. The detailed results are provided in Fig. 6. From these plots one can confirm that: (i) Our DOCO algorithm achieves better performance than D-OCG under the same information feedback (of course, D-OCG exhibits no constraint violations); and (ii) For both algorithms, the performance is degraded from full information to bandit feedback.

## 5 CONCLUSIONS

In this paper, we consider the distributed online convex optimization problem with long-term constraints under full-information and bandit feedback. By introducing and exploiting the notion of online augmented Lagrangian function, we develop distributed algorithms that are based on consensus algorithms. For the case of full-information feedback, we establish sub-linear regret and cumulative absolute constraint violations that match those of centralized online optimization in the literature. Moreover, we also establish sub-linear regret and constraint violation in the case of bandit feedback, where the loss function can be locally evaluated at one point in each time-step.

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

## APPENDIX

## A  PROOF OF THEOREM 1

### A.1  KEY LEMMAS

The following two lemmas are crucial to the convergence analysis of Algorithm 1. The first lemma establishes the basic convergence results of Algorithm 1.

**Lemma 1 (Basic Convergence)** *Let Assumptions 1 and 2 hold. For every node $i^\bullet \in \mathcal{V}$ and $T \geq 1$, we have*

$$
\text{Reg}(i^\bullet, T) \leq \sum_{t=1}^{T} \frac{\sum_{i=1}^{N} \|\mathbf{x}_i(t) - \mathbf{x}^\star\|^2 - \sum_{i=1}^{N} \|\mathbf{x}_i(t+1) - \mathbf{x}^\star\|^2}{2\beta_t} + NG^2 \sum_{t=1}^{T} \beta_t
$$

$$
+ G \sum_{t=1}^{T} \sum_{i=1}^{N} \|\mathbf{x}_{i^\bullet}(t) - \mathbf{x}_i(t)\| - \sum_{t=1}^{T} \sum_{i=1}^{N} \sum_{s=1}^{p} [c_s(\mathbf{x}_i(t))]_+^2 \left( \frac{1}{\eta_{t-1}} - pG^2 \frac{\beta_t}{\eta_{t-1}^2} \right).
$$

*Proof.* To simplify the presentation, we write

$$
\nabla_i(t) \triangleq \nabla_{\mathbf{x}} \mathsf{L}_{i,t}(\mathbf{x}_i(t), \boldsymbol{\lambda}_i(t)).
$$

We study the general evolution of $\|\mathbf{x}_i(t+1) - \mathbf{x}^\star\|^2$,

$$
\|\mathbf{x}_i(t+1) - \mathbf{x}^\star\|^2 = \|\Pi_{\mathcal{B}}(\mathbf{p}_i(t)) - \mathbf{x}^\star\|^2 \leq \|\mathbf{p}_i(t) - \mathbf{x}^\star\|^2
$$

where the inequality is based on the non-expansiveness of the Euclidean projection and $\mathbf{x}^\star \in \mathcal{X} \subseteq \mathcal{B}$. Expanding the right-hand side, we further obtain

$$
\sum_{i=1}^{N} \|\mathbf{x}_i(t+1) - \mathbf{x}^\star\|^2 \leq \sum_{i=1}^{N} \left\| \sum_{j=1}^{N} [\mathbf{A}(t)]_{ij} [\mathbf{x}_j(t) - \beta_t \nabla_j(t)] - \mathbf{x}^\star \right\|^2
$$

$$
\leq \sum_{i=1}^{N} \sum_{j=1}^{N} [\mathbf{A}(t)]_{ij} \|\mathbf{x}_j(t) - \mathbf{x}^\star - \beta_t \nabla_j(t)\|^2
$$

$$
\leq \sum_{i=1}^{N} \|\mathbf{x}_i(t) - \mathbf{x}^\star - \beta_t \nabla_i(t)\|^2
$$

$$
= \sum_{i=1}^{N} \|\mathbf{x}_i(t) - \mathbf{x}^\star\|^2 + \beta_t^2 \sum_{i=1}^{N} \|\nabla_i(t)\|^2 - 2\beta_t \sum_{i=1}^{N} \nabla_i(t)^\mathsf{T} (\mathbf{x}_i(t) - \mathbf{x}^\star)
$$

$$
\leq \sum_{i=1}^{N} \|\mathbf{x}_i(t) - \mathbf{x}^\star\|^2 + \beta_t^2 \sum_{i=1}^{N} \|\nabla_i(t)\|^2
$$

$$
- 2\beta_t \sum_{i=1}^{N} [\mathsf{L}_{i,t}(\mathbf{x}_i(t), \boldsymbol{\lambda}_i(t)) - \mathsf{L}_{i,t}(\mathbf{x}^\star, \boldsymbol{\lambda}_i(t))]
$$

$$(7)$$

where the second and third inequalities follow from the doubly stochasticity of $\mathbf{A}(t)$ and the last inequality from the convexity of $\mathsf{L}_{i,t}(\mathbf{x}, \boldsymbol{\lambda})$ with respect to $\mathbf{x}$. Combining the preceding inequality with the definition of online augmented Lagrangian function in (3), yields

$$
\sum_{i=1}^{N} \left[ \ell_{i,t}(\mathbf{x}_i(t)) + \sum_{s=1}^{p} [\boldsymbol{\lambda}_i(t)]_s [c_s(\mathbf{x}_i(t))]_+ - \frac{\eta_t}{2} \|\boldsymbol{\lambda}_i(t)\|^2 \right.
$$

$$
\left. - \left( \ell_{i,t}(\mathbf{x}^\star) + \sum_{s=1}^{p} [\boldsymbol{\lambda}_i(t)]_s [c_s(\mathbf{x}^\star)]_+ - \frac{\eta_t}{2} \|\boldsymbol{\lambda}_i(t)\|^2 \right) \right]
$$

$$
\leq \frac{\sum_{i=1}^{N} \|\mathbf{x}_i(t) - \mathbf{x}^\star\|^2 - \sum_{i=1}^{N} \|\mathbf{x}_i(t+1) - \mathbf{x}^\star\|^2}{2\beta_t} + \frac{\beta_t}{2} \sum_{i=1}^{N} \|\nabla_i(t)\|^2. \tag{8}
$$

On the other hand, it follows from Assumption 2 that

$$
\|\nabla_i(t)\|^2 = \left\|\nabla\ell_{i,t}(\mathbf{x}_i(t)) + \sum_{s=1}^{p}[\boldsymbol{\lambda}_i(t)]_s\partial[c_s(\mathbf{x}_i(t))]_+\right\|^2
$$

$$
\leq 2\|\nabla\ell_{i,t}(\mathbf{x}_i(t))\|^2 + 2p\sum_{s=1}^{p}[\boldsymbol{\lambda}_i(t)]_s^2\|\partial[c_s(\mathbf{x}_i(t))]_+\|^2
$$

$$
\leq 2G^2 + 2pG^2\sum_{s=1}^{p}[\boldsymbol{\lambda}_i(t)]_s^2
$$

$$
= 2G^2 + 2pG^2\sum_{s=1}^{p}\frac{[c_s(\mathbf{x}_i(t))]_+^2}{\eta_{t-1}^2} \tag{9}
$$

where the last equality follows from the dual update (4). Combining the inequalities (8) and (9), and using the fact that $[\boldsymbol{\lambda}_i(t)]_s = \frac{[c_s(\mathbf{x}_i(t))]_+}{\eta_{t-1}}$ (cf. (4)), we obtain

$$
\sum_{i=1}^{N}[\ell_{i,t}(\mathbf{x}_i(t)) - \ell_{i,t}(\mathbf{x}^\star)] \leq \frac{\sum_{i=1}^{N}\|\mathbf{x}_i(t) - \mathbf{x}^\star\|^2 - \sum_{i=1}^{N}\|\mathbf{x}_i(t+1) - \mathbf{x}^\star\|^2}{2\beta_t} + NG^2\beta_t
$$

$$
- \sum_{i=1}^{N}\sum_{s=1}^{p}[c_s(\mathbf{x}_i(t))]_+^2\left(\frac{1}{\eta_{t-1}} - pG^2\frac{\beta_t}{\eta_{t-1}^2}\right). \tag{10}
$$

The left-hand side can be further lower bounded by

$$
\ell_{i,t}(\mathbf{x}_i(t)) = \ell_{i,t}(\mathbf{x}_{i^\bullet}(t)) + \ell_{i,t}(\mathbf{x}_i(t)) - \ell_{i,t}(\mathbf{x}_{i^\bullet}(t)) \geq \ell_{i,t}(\mathbf{x}_{i^\bullet}(t)) - G\|\mathbf{x}_i(t) - \mathbf{x}_{i^\bullet}(t)\| \tag{11}
$$

summing the inequalities in (10) over $t = 1,\ldots,T$, and using the bound (11) and definition of regret (1), we arrive at the desired conclusion. $\qquad\square$

**Remark 1** *The first two terms in Lemma 1 are optimization errors that are common in the analysis of online optimization algorithms, the third term is the cost of aligning the decisions of nodes, and the last term is the penalty incurred by the violation of constraints.*

The second lemma establishes a bound on the disagreement among all the nodes, which is measured by the difference between the norms of decisions of nodes.

**Lemma 2 (Disagreement)** *Let Assumptions 1–4 hold. For every node $i^\bullet \in \mathcal{V}$ and $T \geq 1$, we have*

$$
\sum_{t=1}^{T}\sum_{i=1}^{N}\|\mathbf{x}_{i^\bullet}(t) - \mathbf{x}_i(t)\| \leq N\hat{C}G\sum_{t=1}^{T}\beta_t + \hat{C}G\sum_{t=1}^{T}\sum_{i=1}^{N}\sum_{s=1}^{p}[c_s(\mathbf{x}_i(t))]_+\frac{\beta_t}{\eta_{t-1}}
$$

*where $\hat{C}$ is given in Theorem 1.*

*Proof.* By deriving the general expressions for the average decision $\bar{\mathbf{x}}(t+1) = \frac{1}{N}\sum_{i=1}^{N}\mathbf{x}_i(t+1)$ and $\mathbf{x}_i(t+1)$ it follows that

$$
\sum_{i=1}^{N}\|\bar{\mathbf{x}}(t+1) - \mathbf{x}_i(t+1)\| \leq \sum_{i=1}^{N}\sum_{m=1}^{t}\beta_m\sum_{j=1}^{N}\left|[\mathbf{A}(t,m)]_{i,j} - N^{-1}\right|\|\nabla_j(m)\|
$$

$$
+ \sum_{i=1}^{N}\sum_{m=1}^{t-1}\sum_{j=1}^{N}\left|[\mathbf{A}(t,m+1)]_{i,j} - N^{-1}\right|\cdot\|\Pi_{\mathcal{B}}(\mathbf{p}_j(m)) - \mathbf{p}_j(m)\|
$$

$$
+ 2\sum_{i=1}^{N}\|\Pi_{\mathcal{B}}(\mathbf{p}_i(t)) - \mathbf{p}_i(t)\|
$$

$$
\tag{12}
$$

where $\mathbf{A}(t,m) = \mathbf{A}(t)\cdots\mathbf{A}(m), \forall t \geq m \geq 1$ and $\mathbf{A}(t,t) = \mathbf{A}(t)$. On the other hand, by resorting to Corollary 1 in Nedic et al. (2008), we have that, for all $t \geq m \geq 1$,

$$\left|[\mathbf{A}(t,m)]_{i,j} - N^{-1}\right| \leq \left(1 - \frac{\zeta}{4N^2}\right)^{\frac{t-m}{B}-2}. \tag{13}$$

Combining the inequalities (12) and (13), we obtain

$$\sum_{t=1}^{T}\sum_{i=1}^{N}\|\bar{\mathbf{x}}(t) - \mathbf{x}_i(t)\| \leq \left(\frac{3N}{\psi^{2+1/B}(1-\psi^{1/B})} + 4\right)\sum_{t=1}^{T-1}\beta_t\sum_{i=1}^{N}\|\nabla_i(t)\|. \tag{14}$$

Then, combining (9) and (14) with the following inequality,

$$\sum_{i=1}^{N}\|\mathbf{x}_{i^\bullet}(t) - \mathbf{x}_i(t)\| = \sum_{i=1}^{N}\|\mathbf{x}_{i^\bullet}(t) - \bar{\mathbf{x}}(t) + \bar{\mathbf{x}}(t) - \mathbf{x}_i(t)\|$$

$$= \sum_{i=1}^{N}\|\mathbf{x}_{i^\bullet}(t) - \bar{\mathbf{x}}(t)\| + \sum_{i=1}^{N}\|\bar{\mathbf{x}}(t) - \mathbf{x}_i(t)\|$$

$$\leq (N+1)\sum_{i=1}^{N}\|\bar{\mathbf{x}}(t) - \mathbf{x}_i(t)\|$$

we arrive at the conclusion. The proof is complete. $\qquad\square$

## A.2 PROOF OF THE THEOREM

We consider an arbitrary unit $i^\bullet \in \mathcal{V}$ and establish a regret bound for that unit. Combining the results in Lemmas 1 and 2, we have

$$\text{Reg}(i^\bullet, T) \leq \sum_{t=1}^{T}\frac{\sum_{i=1}^{N}\|\mathbf{x}_i(t) - \mathbf{x}^\star\|^2 - \sum_{i=1}^{N}\|\mathbf{x}_i(t+1) - \mathbf{x}^\star\|^2}{2\beta_t} + (1+\hat{C})NG^2\sum_{t=1}^{T}\beta_t$$

$$+ \hat{C}G^2\sum_{t=1}^{T}\sum_{i=1}^{N}\sum_{s=1}^{p}[c_s(\mathbf{x}_i(t))]_+\frac{\beta_t}{\eta_{t-1}}$$

$$- \sum_{t=1}^{T}\sum_{i=1}^{N}\sum_{s=1}^{p}[c_s(\mathbf{x}_i(t))]_+^2\left(\frac{1}{\eta_{t-1}} - pG^2\frac{\beta_t}{\eta_{t-1}^2}\right). \tag{15}$$

Substituting $\eta_t = \frac{1}{T^c}$ and $\beta_t = \frac{1}{apG^2T^c}$ into the preceding inequality, yields

$$\text{Reg}(i^\bullet, T) \leq \frac{1}{2}apG^2\left(\sum_{i=1}^{N}\|\mathbf{x}_i(1) - \mathbf{x}^\star\|^2\right)T^c + \frac{1}{ap}N(1+\hat{C})T^{1-c}$$

$$+ \frac{1}{ap}\hat{C}\sum_{t=1}^{T}\sum_{i=1}^{N}\sum_{s=1}^{p}[c_s(\mathbf{x}_i(t))]_+ - \left(1 - \frac{1}{a}\right)T^c\sum_{t=1}^{T}\sum_{i=1}^{N}\sum_{s=1}^{p}[c_s(\mathbf{x}_i(t))]_+^2. \tag{16}$$

We turn our attention to bound the last two terms. To this end, write

$$\rho \triangleq \sum_{t=1}^{T}\sum_{i=1}^{N}\sum_{s=1}^{p}[c_s(\mathbf{x}_i(t))]_+ \tag{17}$$

which implies that

$$\sum_{t=1}^{T}\sum_{i=1}^{N}\sum_{s=1}^{p}[c_s(\mathbf{x}_i(t))]_+^2 \geq \frac{1}{pNT}\left(\sum_{t=1}^{T}\sum_{i=1}^{N}\sum_{s=1}^{p}[c_s(\mathbf{x}_i(t))]_+\right)^2 = \frac{1}{pNT}\rho^2 \tag{18}$$

because of the inequality that $(a_1 + \cdots + a_n)^2 \leq n\sum_{i=1}^{n}a_i^2$. Hence, (16) now becomes

$$\text{Reg}(i^\bullet, T) \leq \frac{1}{2}apNG^2R_\mathcal{X}^2T^c + \frac{1}{ap}N(1+\hat{C})T^{1-c} + \underbrace{\frac{1}{ap}\hat{C}\rho - \left(1 - \frac{1}{a}\right)\frac{1}{pNT^{1-c}}\rho^2}_{\triangleq f(\rho)}. \tag{19}$$

where we have used Assumption 1, i.e., $\sum_{i=1}^{N} \|\mathbf{x}_i(1) - \mathbf{x}^\star\|^2 = \sum_{i=1}^{N} \|\mathbf{x}^\star\|^2 \leq NR_\mathcal{X}^2$. We can replace the term $f(\rho)$ by the following,

$$\max_{\rho \geq 0} f(\rho) = \frac{\left(\frac{\hat{C}}{ap}\right)^2}{4\left(1 - \frac{1}{a}\right)\frac{1}{pNT^{1-c}}} = \frac{N\hat{C}^2}{4a(a-1)p}T^{1-c}. \tag{20}$$

Hence, combining the inequalities (19) and (20), we finally have

$$\mathsf{Reg}(i^\bullet, T) \leq \frac{1}{2}apNG^2R_\mathcal{X}^2 T^c + \frac{1}{ap}N(1+\hat{C})T^{1-c} + \max_{\rho \geq 0} f(\rho)$$

$$\leq \frac{1}{2}apNG^2R_\mathcal{X}^2 T^c + \left(\frac{1}{ap}N(1+\hat{C}) + \frac{N\hat{C}^2}{4a(a-1)p}\right)T^{1-c}. \tag{21}$$

Therefore, we complete the first statement of Theorem 1.

We are left to bound the CACV. From (10) it follows that

$$\sum_{t=1}^{T}\sum_{i=1}^{N}[\ell_{i,t}(\mathbf{x}_i(t)) - \ell_{i,t}(\mathbf{x}^\star)] \leq \sum_{t=1}^{T} \frac{\sum_{i=1}^{N}\|\mathbf{x}_i(t) - \mathbf{x}^\star\|^2 - \sum_{i=1}^{N}\|\mathbf{x}_i(t+1) - \mathbf{x}^\star\|^2}{2\beta_t}$$

$$- \sum_{t=1}^{T}\sum_{i=1}^{N}\sum_{s=1}^{p}[c_s(\mathbf{x}_i(t))]_+^2\left(\frac{1}{\eta_{t-1}} - pG^2\frac{\beta_t}{\eta_{t-1}^2}\right)$$

$$+ NG^2\sum_{t=1}^{T}\beta_t \tag{22}$$

the right-hand side can be bounded by following similar lines as that of the regret analysis, that is,

$$\text{r.h.s. of (22)} \leq \frac{1}{2}apNG^2R_\mathcal{X}^2 T^c + \frac{1}{ap}NT^{1-c} - \left(1 - \frac{1}{a}\right)T^c\sum_{t=1}^{T}\sum_{i=1}^{N}\sum_{s=1}^{p}[c_s(\mathbf{x}_i(t))]_+^2 \tag{23}$$

the left-hand side on (22) can be bounded as follows, according to Assumptions 1 and 2:

$$\text{l.h.s. of (22)} \geq -G\sum_{t=1}^{T}\sum_{i=1}^{N}\|\mathbf{x}_i(t) - \mathbf{x}^\star\| \geq -2NGR_\mathcal{X}T. \tag{24}$$

Combining (23) and (24) and regrouping terms, we have

$$\sum_{t=1}^{T}\sum_{i=1}^{N}\sum_{s=1}^{p}[c_s(\mathbf{x}_i(t))]_+^2 \leq \frac{a^2p}{2(a-1)}NG^2R_\mathcal{X}^2 + \frac{N}{(a-1)p}T^{1-2c} + \frac{2a}{a-1}NGR_\mathcal{X}T^{1-c}$$

$$\leq \frac{N}{a-1}\left(\frac{1}{p} + 2aGR_\mathcal{X} + \frac{1}{2}a^2pG^2R_\mathcal{X}^2\right)T^{1-c}$$

the desired bound follows by combining the preceding inequality and (18). The proof is complete.

## B  PROOF OF THEOREM 2

We first derive the bound on CACV. Note that $\mathsf{L}_{i,t}(\mathbf{x}, \boldsymbol{\lambda})$ is $\sigma$-strongly convex, according to Assumption 5. This fact, combined with (7), leads to

$$\sum_{i=1}^{N}\|\mathbf{x}_i(t+1) - \mathbf{x}^\star\|^2 \leq \sum_{i=1}^{N}\|\mathbf{x}_i(t) - \mathbf{x}^\star\|^2 + \beta_t^2\sum_{i=1}^{N}\|\nabla_i(t)\|^2$$

$$- 2\beta_t\sum_{i=1}^{N}\left[\mathsf{L}_{i,t}(\mathbf{x}_i(t), \boldsymbol{\lambda}_i(t)) - \mathsf{L}_{i,t}(\mathbf{x}^\star, \boldsymbol{\lambda}_i(t)) + \frac{\sigma}{2}\|\mathbf{x}_i(t) - \mathbf{x}^\star\|^2\right] \tag{25}$$

regrouping the terms, we further have

$$\sum_{t=1}^{T}\sum_{i=1}^{N}[\mathsf{L}_{i,t}(\mathbf{x}_i(t),\boldsymbol{\lambda}_i(t)) - \mathsf{L}_{i,t}(\mathbf{x}^\star,\boldsymbol{\lambda}_i(t))]$$

$$\leq \sum_{t=1}^{T}\frac{\sum_{i=1}^{N}\|\mathbf{x}_i(t)-\mathbf{x}^\star\|^2 - \sum_{i=1}^{N}\|\mathbf{x}_i(t+1)-\mathbf{x}^\star\|^2}{2\beta_t}$$

$$- \frac{\sigma}{2}\sum_{t=1}^{T}\|\mathbf{x}_i(t)-\mathbf{x}^\star\|^2 + \sum_{t=1}^{T}\frac{\beta_t}{2}\sum_{i=1}^{N}\|\nabla_i(t)\|^2$$

$$= \frac{1}{2}\sum_{t=2}^{T}\left(\frac{1}{\beta_t}-\frac{1}{\beta_{t-1}}-\sigma\right)\sum_{i=1}^{N}\|\mathbf{x}_i(t)-\mathbf{x}^\star\|^2 + \frac{1}{2}\left(\frac{1}{\beta_1}-\sigma\right)\sum_{i=1}^{N}\|\mathbf{x}_i(1)-\mathbf{x}^\star\|^2$$

$$- \frac{1}{2\beta_T}\sum_{i=1}^{N}\|\mathbf{x}_i(T+1)-\mathbf{x}^\star\|^2 + \sum_{t=1}^{T}\frac{\beta_t}{2}\sum_{i=1}^{N}\|\nabla_i(t)\|^2.$$

Substituting $\beta_t = \frac{1}{\sigma t}$ into the preceding inequality and dropping the negative term, yields

$$\sum_{t=1}^{T}\sum_{i=1}^{N}[\mathsf{L}_{i,t}(\mathbf{x}_i(t),\boldsymbol{\lambda}_i(t)) - \mathsf{L}_{i,t}(\mathbf{x}^\star,\boldsymbol{\lambda}_i(t))] \leq \sum_{t=1}^{T}\frac{\beta_t}{2}\sum_{i=1}^{N}\|\nabla_i(t)\|^2. \tag{26}$$

Noting that $\boldsymbol{\lambda}_i(t)$ is the maximizer of $\mathsf{L}_{i,t}((\mathbf{x}_i(t),\boldsymbol{\lambda})$ over $\boldsymbol{\lambda} \in \mathbb{R}_+^p$, i.e., $\mathsf{L}_{i,t}(\mathbf{x}_i(t),\boldsymbol{\lambda}_i(t)) \geq \mathsf{L}_{i,t}(\mathbf{x}_i(t),\boldsymbol{\lambda})$ for all $\boldsymbol{\lambda} \in \mathbb{R}_+^p$, we have the following estimate for all $\boldsymbol{\lambda} \in \mathbb{R}_+^p$, according to (26),

$$\sum_{t=1}^{T}\sum_{i=1}^{N}[\mathsf{L}_{i,t}(\mathbf{x}_i(t),\boldsymbol{\lambda}) - \mathsf{L}_{i,t}(\mathbf{x}^\star,\boldsymbol{\lambda}_i(t))] \leq \sum_{t=1}^{T}\frac{\beta_t}{2}\sum_{i=1}^{N}\|\nabla_i(t)\|^2. \tag{27}$$

Expanding the left-hand side by using the definition of online augmented Lagrangian function in (3), we further have

$$\sum_{t=1}^{T}\sum_{i=1}^{N}\left[\ell_{i,t}(\mathbf{x}_i(t)) + \sum_{s=1}^{p}[\boldsymbol{\lambda}]_s[c_s(\mathbf{x}_i(t))]_+ - \frac{\eta_t}{2}\|\boldsymbol{\lambda}\|^2\right.$$

$$\left. - \left(\ell_{i,t}(\mathbf{x}^\star) + \sum_{s=1}^{p}[\boldsymbol{\lambda}_i(t)]_s[c_s(\mathbf{x}^\star)]_+ - \frac{\eta_t}{2}\|\boldsymbol{\lambda}_i(t)\|^2\right)\right]$$

$$\leq \sum_{t=1}^{T}\frac{\beta_t}{2}\sum_{i=1}^{N}\|\nabla_i(t)\|^2 \leq NG^2\sum_{t=1}^{T}\beta_t + pG^2\sum_{t=1}^{T}\sum_{i=1}^{N}\beta_t\|\boldsymbol{\lambda}_i(t)\|^2 \tag{28}$$

where we recalled (9). Applying $\eta_t = 2pG^2\beta_t$ to (28), we find that the last terms on both sides will cancel each other out. This leads to

$$\sum_{t=1}^{T}\sum_{i=1}^{N}[\ell_{i,t}(\mathbf{x}_i(t)) - \ell_{i,t}(\mathbf{x}^\star)] + \underbrace{\sum_{s=1}^{p}\left(\sum_{t=1}^{T}\sum_{i=1}^{N}[c(\mathbf{x}_i(t))]_+\right)[\boldsymbol{\lambda}]_s - \sum_{s=1}^{p}\left(\frac{1}{2}N\sum_{t=1}^{T}\eta_t\right)[\boldsymbol{\lambda}]_s^2}_{\triangleq g(\boldsymbol{\lambda})}$$

$$\leq NG^2\sum_{t=1}^{T}\beta_t \tag{29}$$

note that (29) holds for all $\boldsymbol{\lambda} \in \mathbb{R}_+^d$, and hence we can replace $g(\boldsymbol{\lambda})$ by the following,

$$\max_{\boldsymbol{\lambda}\in\mathbb{R}_+^p} g(\boldsymbol{\lambda}) \leq \sum_{t=1}^{T}\sum_{i=1}^{N}[\ell_{i,t}(\mathbf{x}^\star) - \ell_{i,t}(\mathbf{x}_i(t))] + NG^2\sum_{t=1}^{T}\beta_t \leq 2NGR_\mathcal{X}T + NG^2\sum_{t=1}^{T}\beta_t \tag{30}$$

where the last inequality follows from the same reasoning as that of (24). For the left-hand side on (30), we have

$$\max_{\boldsymbol{\lambda}\in\mathbb{R}_+^d} g(\boldsymbol{\lambda}) = \sum_{s=1}^{p}\frac{\left(\sum_{t=1}^{T}\sum_{i=1}^{N}[c_s(\mathbf{x}_i(t))]_+\right)^2}{2N\sum_{t=1}^{T}\eta_t} \geq \frac{\left(\sum_{t=1}^{T}\sum_{i=1}^{N}\sum_{s=1}^{p}[c_s(\mathbf{x}_i(t))]_+\right)^2}{2pN\sum_{t=1}^{T}\eta_t} \tag{31}$$

Combining the inequalities (30) and (31) with the following estimate,

$$\sum_{t=1}^{T} \frac{1}{t} = 1 + \sum_{t=2}^{T} \frac{1}{t} \leq 1 + \int_{1}^{T} \frac{1}{u} \mathrm{d}u = 1 + \log(T) \tag{32}$$

we further obtain for all $T \geq 3$,

$$\sum_{t=1}^{T} \sum_{i=1}^{N} \sum_{s=1}^{p} [c_s(\mathbf{x}_i(t))]_+ \leq \sqrt{\frac{16p^2 N^2 G^3 R_{\mathcal{X}}}{\sigma} T \log(T) + \frac{16p^2 N^2 G^4}{\sigma^2} \log^2(T)}$$

$$\leq \left( \frac{4pNG^{3/2}\sqrt{R_{\mathcal{X}}}}{\sqrt{\sigma}} + \frac{4pNG^2}{\sigma} \right) \sqrt{T \log(T)}. \tag{33}$$

Next, we turn our attention to the first statement, i.e., the regret bound. Again We consider an arbitrary unit $i^{\bullet} \in \mathcal{V}$. Combining the inequalities (30) and (31), and using the notation (17), we obtain

$$NG^2 \sum_{t=1}^{T} \beta_t - \frac{1}{2pN \sum_{t=1}^{T} \eta_t} \rho^2 \geq \sum_{t=1}^{T} \sum_{i=1}^{N} [\ell_{i,t}(\mathbf{x}_i(t)) - \ell_{i,t}(\mathbf{x}^{\star})]$$

$$\geq \mathsf{Reg}(i^{\bullet}, T) - G \sum_{t=1}^{T} \sum_{i=1}^{N} \|\mathbf{x}_{i^{\bullet}}(t) - \mathbf{x}_i(t)\|. \tag{34}$$

Applying Lemma 2 to the preceding inequality gives

$$\mathsf{Reg}(i^{\bullet}, T) \leq NG^2(1 + \hat{C}) \sum_{t=1}^{T} \beta_t - \frac{1}{2pN \sum_{t=1}^{T} \eta_t} \rho^2 + \hat{C}G^2 \sum_{t=1}^{T} \sum_{i=1}^{N} \sum_{s=1}^{p} [c_s(\mathbf{x}_i(t))]_+ \frac{\beta_t}{\eta_{t-1}} \tag{35}$$

substituting the expressions for $\beta_t$ and $\eta_t$ into (35), we have

$$\mathsf{Reg}(i^{\bullet}, T) \leq NG^2(1 + \hat{C}) \sum_{t=1}^{T} \beta_t + \underbrace{\frac{1}{2p}\hat{C}\rho - \frac{1}{2pN \sum_{t=1}^{T} \eta_t} \rho^2}_{\triangleq h(\rho)}. \tag{36}$$

Then, following an argument similar to that of (20) and (21), we get

$$\mathsf{Reg}(i^{\bullet}, T) \leq NG^2(1 + \hat{C}) \sum_{t=1}^{T} \beta_t + \frac{(\frac{1}{2p}\hat{C})^2}{4 \frac{1}{2pN \sum_{t=1}^{T} \eta_t}} \leq NG^2(1 + \hat{C}) \sum_{t=1}^{T} \beta_t + \frac{1}{8p} N\hat{C}^2 \sum_{t=1}^{T} \eta_t$$

$$\leq \left( \frac{2NG^2(1 + \hat{C})}{\sigma} + \frac{N\hat{C}^2 G^2}{2\sigma} \right) \log(T). \tag{37}$$

The proof is complete.

## C  PROOF OF THEOREM 3

### C.1  KEY LEMMA

The proof relies on the properties of the one-point gradient estimator $\tilde{\nabla}\ell_{i,t}(\mathbf{x}_i(t))$, which can be viewed as a distributed version of the one in Flaxman et al. (2005). In particular, we have the following lemma that characterizes the properties of the one-point gradient estimator and the relation between $\ell_{i,t}(\mathbf{x})$ and its smoothed version $\tilde{\ell}_{i,t}(\mathbf{x}; \varepsilon)$.

**Lemma 3**    *(i) Let $G$ be the uniform Lipschitz constant of the loss functions $\ell_{i,t}(\mathbf{x})$ over $\mathcal{B}$, then the smoothed loss functions $\tilde{\ell}_{i,t}(\mathbf{x})$ are Lipschitz continuous with the same constant $G$ and we have that, for all $\mathbf{x} \in \mathcal{B}$,*

$$\left| \tilde{\ell}_{i,t}(\mathbf{x}; \varepsilon) - \ell_{i,t}(\mathbf{x}) \right| \leq G\varepsilon.$$

(ii) *The one-point gradient estimator satisfies*

$$\mathbb{E}\left[\tilde{\nabla}\ell_{i,t}(\mathbf{x}_i(t))\right] = \nabla\tilde{\ell}_{i,t}(\mathbf{x}_i(t); \varepsilon_t).$$

(iii) *Let Assumption 6 hold, then the one-point gradient estimator satisfies*

$$\left\|\tilde{\nabla}\ell_{i,t}(\mathbf{x}_i(t))\right\| \leq \frac{Cd}{\varepsilon_t}.$$

## C.2 PROOF OF THE THEOREM

Denote

$$\tilde{\nabla}_i(t) \triangleq \tilde{\nabla}\ell_{i,t}(\mathbf{x}_i(t)) + \sum_{s=1}^{p}[\boldsymbol{\lambda}_i(t)]_s\partial[c_s(\mathbf{x}_i(t))]_+ \tag{38}$$

then, it follows from Lemma 3(ii) that

$$\mathbb{E}\left[\tilde{\nabla}_i(t)\right] = \nabla\tilde{\ell}_{i,t}(\mathbf{x}_i(t); \varepsilon_t) + \sum_{s=1}^{p}[\boldsymbol{\lambda}_i(t)]_s\partial[c_s(\mathbf{x}_i(t))]_+ = \nabla_{\mathbf{x}}\tilde{L}_{i,t}(\mathbf{x}_i(t), \boldsymbol{\lambda}_i(t)). \tag{39}$$

Following similar lines as that of Lemma 1, we immediately have

$$\sum_{i=1}^{N}\|\mathbf{x}_i(t+1) - \mathbf{x}_\pi^\star\|^2 = \sum_{i=1}^{N}\|\mathbf{x}_i(t) - \mathbf{x}_\pi^\star\|^2 + \beta_t^2\sum_{i=1}^{N}\left\|\tilde{\nabla}_i(t)\right\|^2 - 2\beta_t\sum_{i=1}^{N}\tilde{\nabla}_i(t)^\mathsf{T}(\mathbf{x}_i(t) - \mathbf{x}_\pi^\star) \tag{40}$$

where $\mathbf{x}_\pi^\star = (1-\pi)\mathbf{x}^\star \in (1-\pi)\mathcal{X} \subseteq (1-\pi)\mathcal{B}$. Taking expectation on both sides of (40) and using (39), yields

$$\sum_{i=1}^{N}\mathbb{E}\left[\tilde{L}_{i,t}(\mathbf{x}_i(t), \boldsymbol{\lambda}_i(t)) - \tilde{L}_{i,t}(\mathbf{x}^\star, \boldsymbol{\lambda}_i(t))\right]$$

$$\leq \frac{\sum_{i=1}^{N}\mathbb{E}[\|\mathbf{x}_i(t) - \mathbf{x}_\pi^\star\|^2] - \sum_{i=1}^{N}\mathbb{E}[\|\mathbf{x}_i(t+1) - \mathbf{x}_\pi^\star\|^2]}{2\beta_t} + \frac{\beta_t}{2}\sum_{i=1}^{N}\mathbb{E}\left[\left\|\tilde{\nabla}_i(t)\right\|^2\right]. \tag{41}$$

Using the definition (5), the left-hand side on (41) becomes

$$\sum_{i=1}^{N}\mathbb{E}\left[\tilde{\ell}_{i,t}(\mathbf{x}_i(t); \varepsilon_t) + \sum_{s=1}^{p}[\boldsymbol{\lambda}_i(t)]_s[c_s(\mathbf{x}_i(t))]_+ - \frac{\eta_t}{2}\|\boldsymbol{\lambda}_i(t)\|^2\right.$$

$$\left. - \left(\tilde{\ell}_{i,t}(\mathbf{x}_\pi^\star; \varepsilon_t) + \sum_{s=1}^{p}[\boldsymbol{\lambda}_i(t)]_s[c_s(\mathbf{x}_\pi^\star)]_+ - \frac{\eta_t}{2}\|\boldsymbol{\lambda}_i(t)\|^2\right)\right]$$

$$= \sum_{i=1}^{N}\mathbb{E}\left[\tilde{\ell}_{i,t}(\mathbf{x}_i(t); \varepsilon_t) - \tilde{\ell}_{i,t}(\mathbf{x}_\pi^\star; \varepsilon_t)\right] + \sum_{i=1}^{N}\sum_{s=1}^{p}\frac{\mathbb{E}\left[[c_s(\mathbf{x}_i(t))]_+^2\right]}{\eta_{t-1}} \tag{42}$$

where the equality follows from $[c_s(\mathbf{x}_\pi^\star)]_+ = 0$, because $\mathbf{x}_\pi^\star \in (1-\pi)\mathcal{X} \subset \mathcal{X}$. On the other hand, it follows from (38) and Lemma 3(iii) that

$$\mathbb{E}\left[\left\|\tilde{\nabla}_i(t)\right\|^2\right] = \mathbb{E}\left[\left\|\tilde{\nabla}\ell_{i,t}(\mathbf{x}_i(t)) + \sum_{s=1}^{p}[\boldsymbol{\lambda}_i(t)]_s\partial[c_s(\mathbf{x}_i(t))]_+\right\|^2\right]$$

$$\leq 2\mathbb{E}\left[\|\tilde{\nabla}\ell_{i,t}(\mathbf{x}_i(t))\|^2\right] + 2pG^2\sum_{s=1}^{p}\mathbb{E}\left[[\boldsymbol{\lambda}_i(t)]_s^2\right]$$

$$\leq 2C^2d^2\frac{1}{\varepsilon_t^2} + 2pG^2\sum_{s=1}^{p}\frac{\mathbb{E}\left[[c_s(\mathbf{x}_i(t))]_+^2\right]}{\eta_{t-1}^2} \tag{43}$$

this, combined with equations (41) and (42), gives

$$
\sum_{i=1}^{N} \mathbb{E}\left[\tilde{\ell}_{i,t}(\mathbf{x}_i(t); \varepsilon_t) - \tilde{\ell}_{i,t}(\mathbf{x}_\pi^\star; \varepsilon_t)\right]
$$

$$
\leq NC^2 d^2 \frac{\beta_t}{\varepsilon_t^2} + \frac{\sum_{i=1}^{N} \mathbb{E}\left[\|\mathbf{x}_i(t) - \mathbf{x}_\pi^\star\|^2\right] - \sum_{i=1}^{N} \mathbb{E}\left[\|\mathbf{x}_i(t+1) - \mathbf{x}_\pi^\star\|^2\right]}{2\beta_t}
$$

$$
- \sum_{i=1}^{N} \sum_{s=1}^{p} \mathbb{E}\left[[c_s(\mathbf{x}_i(t))]_+^2\right]\left(\frac{1}{\eta_{t-1}} - pG^2 \frac{\beta_t}{\eta_{t-1}^2}\right). \tag{44}
$$

the left-hand side on (44) can be further lower bounded by utilizing the relation between the losses $\ell_{i,t}$ and their smoothed variants $\tilde{\ell}_{i,t}$ (cf. Lemma 3(i)), given as follows:

$$
\tilde{\ell}_{i,t}(\mathbf{x}_i(t); \varepsilon_t) - \tilde{\ell}_{i,t}(\mathbf{x}_\pi^\star; \varepsilon_t) \geq \ell_{i,t}(\mathbf{x}_i(t)) - \ell_{i,t}(\mathbf{x}_\pi^\star) - 2G\varepsilon_t
$$

$$
\geq \ell_{i,t}(\mathbf{x}_i(t)) - \ell_{i,t}(\mathbf{x}^\star) - GR_\mathcal{X}\pi - 2G\varepsilon_t \tag{45}
$$

where the last inequality follows from the fact that $\ell_{i,t}$ is $G$-Lipschitz and $\|\mathbf{x}^\star\| \leq R_\mathcal{X}$. Summing the inequalities in (44) over $t = 1, \ldots, T$ and using (45), we find that

$$
\sum_{t=1}^{T} \sum_{i=1}^{N} \mathbb{E}\left[\ell_{i,t}(\mathbf{x}_i(t)) - \ell_{i,t}(\mathbf{x}^\star)\right] \leq NGR_\mathcal{X}\pi T + 2NG \sum_{t=1}^{T} \varepsilon_t + NC^2 d^2 \sum_{t=1}^{T} \frac{\beta_t}{\varepsilon_t^2}
$$

$$
+ \sum_{t=1}^{T} \frac{\sum_{i=1}^{N} \mathbb{E}\left[\|\mathbf{x}_i(t) - \mathbf{x}_\pi^\star\|^2\right] - \sum_{i=1}^{N} \mathbb{E}\left[\|\mathbf{x}_i(t+1) - \mathbf{x}_\pi^\star\|^2\right]}{2\beta_t}
$$

$$
- \sum_{t=1}^{T} \sum_{i=1}^{N} \sum_{s=1}^{p} \mathbb{E}\left[[c_s(\mathbf{x}_i(t))]_+^2\right]\left(\frac{1}{\eta_{t-1}} - pG^2 \frac{\beta_t}{\eta_{t-1}^2}\right). \tag{46}
$$

On the other hand, we have the following estimate of the disagreement among nodes by resorting to Lemma 2:

$$
\sum_{t=1}^{T} \sum_{i=1}^{N} \mathbb{E}\left[\|\mathbf{x}_{i\bullet}(t) - \mathbf{x}_i(t)\|\right] \leq \hat{C} \sum_{t=1}^{T-1} \beta_t \sum_{i=1}^{N} \mathbb{E}\left[\left\|\tilde{\nabla}_i(t)\right\|\right]
$$

$$
\leq \hat{C} \sum_{t=1}^{T} \beta_t \sum_{i=1}^{N} \left(\frac{Cd}{\varepsilon_t} + G \sum_{s=1}^{p} \frac{\mathbb{E}\left[[c_s(\mathbf{x}_i(t))]_+\right]}{\eta_{t-1}}\right)
$$

$$
\leq NC\hat{C}d \sum_{t=1}^{T} \frac{\beta_t}{\varepsilon_t} + \hat{C}G \sum_{t=1}^{T} \sum_{i=1}^{N} \sum_{s=1}^{p} \mathbb{E}\left[[c_s(\mathbf{x}_i(t))]_+\right] \frac{\beta_t}{\eta_{t-1}} \tag{47}
$$

where the second inequality is based on (43). Combining inequalities (46) and (47), and following an argument similar to that of Theorem 1, we obtain

$$
\mathbb{E}\left[\text{Reg}(i^\bullet, T)\right] \leq NGR_\mathcal{X}\pi T + 2NG \sum_{t=1}^{T} \varepsilon_t + NC\hat{C}Gd \sum_{t=1}^{T} \frac{\beta_t}{\varepsilon_t} + NC^2 d^2 \sum_{t=1}^{T} \frac{\beta_t}{\varepsilon_t^2}
$$

$$
+ \sum_{t=1}^{T} \frac{\sum_{i=1}^{N} \mathbb{E}\left[\|\mathbf{x}_i(t) - \mathbf{x}_\pi^\star\|^2\right] - \sum_{i=1}^{N} \mathbb{E}\left[\|\mathbf{x}_i(t+1) - \mathbf{x}_\pi^\star\|^2\right]}{2\beta_t}
$$

$$
+ \hat{C}G^2 \sum_{t=1}^{T} \sum_{i=1}^{N} \sum_{s=1}^{p} \mathbb{E}\left[[c_s(\mathbf{x}_i(t))]_+\right] \frac{\beta_t}{\eta_{t-1}}
$$

$$
- \sum_{t=1}^{T} \sum_{i=1}^{N} \sum_{s=1}^{p} \mathbb{E}\left[[c_s(\mathbf{x}_i(t))]_+^2\right]\left(\frac{1}{\eta_{t-1}} - pG^2 \frac{\beta_t}{\eta_{t-1}^2}\right). \tag{48}
$$

Substituting $\eta_t = \frac{1}{T^c}$, $\beta_t = \frac{1}{apG^2T^c}$, $\varepsilon_t = \frac{1}{T^b}$ and $\pi = \frac{1}{R_\mathcal{X}T^b}$ into the preceding inequality, and following similar lines as that of Theorem 1, yields

$$\mathbb{E}\left[\mathrm{Reg}(i^\bullet, T)\right] \leq 3NGT^{1-b} + \frac{NC\hat{C}d}{apG}T^{1+b-c} + \frac{NC^2d^2}{apG^2}T^{1+2b-c}$$

$$+ \frac{1}{2}apNG^2R_\mathcal{X}^2T^c + \frac{N\hat{C}^2}{4a(a-1)p}T^{1-c}. \tag{49}$$

It follows from some simple algebra that the choice of $b = \frac{c}{3}$ yields the optimal regret bound $\mathcal{O}(T^{\max\{1-c/3,c\}})$.

The bound on CACV can be derived by lower bounding the left-hand side on (44), that is,

$$\sum_{i=1}^{N}\mathbb{E}\left[\tilde{\ell}_{i,t}(\mathbf{x}_i(t); \varepsilon_t) - \tilde{\ell}_{i,t}(\mathbf{x}_\pi^\star; \varepsilon_t)\right] \geq -G\sum_{i=1}^{N}\mathbb{E}[\|\mathbf{x}_i(t) - \mathbf{x}_\pi^\star\|] \geq -2NGR_\mathcal{X}$$

where the first inequality follows from Lemma 3(i) and the last one from Assumption 1. This, combined with (44) and the expressions of $\eta_t = \frac{1}{T^c}$, $\beta_t = \frac{1}{apG^2T^c}$, $\varepsilon_t = \frac{1}{T^b}$ and $\pi = \frac{1}{R_\mathcal{X}T^b}$, leads to

$$\sum_{t=1}^{T}\sum_{i=1}^{N}\sum_{s=1}^{p}\mathbb{E}\left[[c_s(\mathbf{x}_i(t))]_+^2\right] \leq \frac{N}{a-1}\left(2aGR_\mathcal{X}T^{1-c} + \frac{1}{2}a^2pG^2R_\mathcal{X}^2 + \frac{C^2d^2}{pG^2}T^{1+2b-2c}\right)$$

$$\leq \frac{N}{a-1}\left(2aGR_\mathcal{X} + \frac{1}{2}a^2pG^2R_\mathcal{X}^2 + \frac{C^2d^2}{pG^2}\right)T^{1-c} \tag{50}$$

where in the last inequality we used $b = \frac{c}{3}$. The desired result follows by combining the preceding inequality with the following,

$$\sum_{t=1}^{T}\sum_{i=1}^{N}\sum_{s=1}^{p}\mathbb{E}\left[[c_s(\mathbf{x}_i(t))]_+\right] \leq \mathbb{E}\left[\left(pNT\sum_{t=1}^{T}\sum_{i=1}^{N}\sum_{s=1}^{p}[c_s(\mathbf{x}_i(t))]_+^2\right)^{1/2}\right]$$

$$\leq \left(pNT\sum_{t=1}^{T}\sum_{i=1}^{N}\sum_{s=1}^{p}\mathbb{E}\left[[c_s(\mathbf{x}_i(t))]_+^2\right]\right)^{1/2} \tag{51}$$

because of Jensen's inequality. The proof is complete.

## D   PROOF OF THEOREM 4

We first claim that the strongly convexity of the loss functions $\ell_{i,t}$ implies the strongly convexity of their smoothed variants $\tilde{\ell}_{i,t}$ with the same constant $\sigma$. This fact leads us to the following bound that is analogous to (26):

$$\sum_{t=1}^{T}\sum_{i=1}^{N}\mathbb{E}\left[\tilde{L}_{i,t}(\mathbf{x}_i(t), \boldsymbol{\lambda}_i(t)) - \tilde{L}_{i,t}(\mathbf{x}^\star, \boldsymbol{\lambda}_i(t))\right] \leq \sum_{t=1}^{T}\frac{\beta_t}{2}\sum_{i=1}^{N}\mathbb{E}\left[\left\|\tilde{\nabla}_i(t)\right\|^2\right]. \tag{52}$$

Following an argument similar to that of Theorem 2, we can replace the left-hand side on (52) by the following, due to the fact that $\boldsymbol{\lambda}_i(t)$ is the maximizer of $\tilde{L}_{i,t}((\mathbf{x}_i(t), \boldsymbol{\lambda})$ over $\boldsymbol{\lambda} \in \mathbb{R}_+^p$:

$$\sum_{t=1}^{T}\sum_{i=1}^{N}\mathbb{E}\left[\tilde{L}_{i,t}(\mathbf{x}_i(t), \boldsymbol{\lambda}) - \tilde{L}_{i,t}(\mathbf{x}_\pi^\star, \boldsymbol{\lambda}_i(t))\right]$$

$$= \sum_{t=1}^{T}\sum_{i=1}^{N}\mathbb{E}\left[\tilde{\ell}_{i,t}(\mathbf{x}_i(t)) + \sum_{s=1}^{p}[\boldsymbol{\lambda}]_s[c_s(\mathbf{x}_i(t))]_+ - \frac{\eta_t}{2}\|\boldsymbol{\lambda}\|^2\right.$$

$$\left. - \left(\tilde{\ell}_{i,t}(\mathbf{x}_\pi^\star) + \sum_{s=1}^{p}[\boldsymbol{\lambda}_i(t)]_s[c_s(\mathbf{x}_\pi^\star)]_+ - \frac{\eta_t}{2}\|\boldsymbol{\lambda}_i(t)\|^2\right)\right] \tag{53}$$

$$= \sum_{t=1}^{T}\sum_{i=1}^{N}\mathbb{E}\left[\tilde{\ell}_{i,t}(\mathbf{x}_i(t)) - \tilde{\ell}_{i,t}(\mathbf{x}_\pi^\star)\right] + \mathbb{E}\left[g(\boldsymbol{\lambda})\right] + \frac{1}{2}\sum_{t=1}^{T}\sum_{i=1}^{N}\eta_t\mathbb{E}\left[\|\boldsymbol{\lambda}_i(t)\|^2\right]$$

where $g(\boldsymbol{\lambda})$ is defined in (29). On the other hand, using (43) we have

$$\mathbb{E}\left[\left\|\tilde{\nabla}_i(t)\right\|^2\right] \le 2C^2 d^2 \frac{1}{\varepsilon_t^2} + 2pG^2 \mathbb{E}\left[\|\boldsymbol{\lambda}_i(t)\|^2\right] \tag{54}$$

Combining the equations (52), (53) and (54), and using $\eta_t = 2pG^2\beta_t$, we find that for all $\boldsymbol{\lambda} \in \mathbb{R}_+^p$ and $T \ge 3$,

$$\mathbb{E}[g(\boldsymbol{\lambda})] \le \sum_{t=1}^{T}\sum_{i=1}^{N} \mathbb{E}\left[\tilde{\ell}_{i,t}(\mathbf{x}_\pi^\star) - \tilde{\ell}_{i,t}(\mathbf{x}_i(t))\right] + NC^2d^2 \sum_{t=1}^{T} \frac{\beta_t}{\varepsilon_t^2}$$

$$\le 2NGR_\mathcal{X}T + \frac{NC^2d^2}{\sigma}T^{2b}\sum_{t=1}^{T}\frac{1}{t}$$

$$\le 2NGR_\mathcal{X}T + \frac{2NC^2d^2}{\sigma}T^{2b}\log(T) \tag{55}$$

where we recalled (32). Substituting $\boldsymbol{\lambda} = \boldsymbol{\lambda}^\star$, the maximizer of $g(\boldsymbol{\lambda})$ over $\boldsymbol{\lambda} \in \mathbb{R}_+^p$, into the (55), the left-hand side on (55) becomes

$$\mathbb{E}[g(\boldsymbol{\lambda}^\star)] = \sum_{s=1}^{p} \frac{\mathbb{E}\left[\left(\sum_{t=1}^{T}\sum_{i=1}^{N}[c_s(\mathbf{x}_i(t))]_+\right)^2\right]}{2N\sum_{t=1}^{T}\eta_t} \ge \sum_{s=1}^{p} \frac{\left(\sum_{t=1}^{T}\sum_{i=1}^{N}\mathbb{E}\left[[c_s(\mathbf{x}_i(t))]_+\right]\right)^2}{2N\sum_{t=1}^{T}\eta_t}$$

$$\ge \frac{\left(\sum_{t=1}^{T}\sum_{i=1}^{N}\sum_{s=1}^{p}\mathbb{E}\left[[c_s(\mathbf{x}_i(t))]_+\right]\right)^2}{2pN\sum_{t=1}^{T}\eta_t} \tag{56}$$

where the first inequality follows from Jensen's inequality. Combining the inequalities in (32), (55), and (56), yields

$$\sum_{t=1}^{T}\sum_{i=1}^{N}\sum_{s=1}^{p}\mathbb{E}\left[[c_s(\mathbf{x}_i(t))]_+\right] \le \frac{4pNG^{3/2}\sqrt{R_\mathcal{X}}}{\sqrt{\sigma}}\sqrt{T\log(T)} + \frac{4pNGCd}{\sigma}T^{1/3}\log(T)$$

where we used $b = \frac{1}{3}$.

We now turn our attention to the regret bound. It follows from (55) and (56) that

$$\sum_{t=1}^{T}\sum_{i=1}^{N}\mathbb{E}\left[\tilde{\ell}_{i,t}(\mathbf{x}_i(t)) - \tilde{\ell}_{i,t}(\mathbf{x}_\pi^\star)\right] \le NC^2d^2 \sum_{t=1}^{T}\frac{\beta_t}{\varepsilon_t^2} - \frac{(\mathbb{E}[\rho])^2}{2pN\sum_{t=1}^{T}\eta_t}$$

this, combined with (45), further leads to

$$\sum_{t=1}^{T}\sum_{i=1}^{N}\mathbb{E}\left[\ell_{i,t}(\mathbf{x}_i(t)) - \ell_{i,t}(\mathbf{x}^\star)\right] \le NGR_\mathcal{X}\pi T + 2NG\sum_{t=1}^{T}\varepsilon_t + NC^2d^2\sum_{t=1}^{T}\frac{\beta_t}{\varepsilon_t^2} - \frac{(\mathbb{E}[\rho])^2}{2pN\sum_{t=1}^{T}\eta_t}. \tag{57}$$

Then, following the similar lines as that of Theorem 2 and using the disagreement estimate (47), we find that

$$\mathbb{E}[\text{Reg}(i^\bullet, T)] \le NGR_\mathcal{X}\pi T + 2NG\sum_{t=1}^{T}\varepsilon_t + NC\hat{C}Gd\sum_{t=1}^{T}\frac{\beta_t}{\varepsilon_t} + NC^2d^2\sum_{t=1}^{T}\frac{\beta_t}{\varepsilon_t^2}$$

$$+ \underbrace{\hat{C}G^2\mathbb{E}[\rho]\frac{\beta_t}{\eta_{t-1}} - \frac{(\mathbb{E}[\rho])^2}{2pN\sum_{t=1}^{T}\eta_t}}_{=h(\mathbb{E}[\rho])}$$

$$\le NGR_\mathcal{X}\pi T + 2NG\sum_{t=1}^{T}\varepsilon_t + NC\hat{C}Gd\sum_{t=1}^{T}\frac{\beta_t}{\varepsilon_t} + NC^2d^2\sum_{t=1}^{T}\frac{\beta_t}{\varepsilon_t^2} + \frac{1}{8p}N\hat{C}^2\sum_{t=1}^{T}\eta_t \tag{58}$$

where the last inequality follows from the same reasoning as that of (35)–(37). This, combined with $\eta_t = \frac{2pG^2}{\sigma t}$, $\beta_t = \frac{1}{\sigma t}$, $\varepsilon_t = \frac{1}{T^b}$ and $\pi = \frac{1}{R_\mathcal{X} T^b}$, leads to

$$\mathbb{E}\left[\mathsf{Reg}(i^\bullet, T)\right] \leq 3NGT^{1-b} + \frac{2NC\hat{C}Gd}{\sigma}T^b \log(T) + \frac{2NC^2d^2}{\sigma}T^{2b} \log(T) + \frac{N\hat{C}^2G^2}{2\sigma} \log(T) \tag{59}$$

hence, the optimal regret bound follows by setting $b = \frac{1}{3}$. The proof is complete.

