# OpenReview forum: "Distributed Online Optimization with Long-Term Constraints"
_ICLR.cc/2020/Conference — Reject_

### Official Review · AnonReviewer2 · 2019-10-22
**Official Blind Review #2**

**Rating:** 6

**Review:**

This paper considers distributed online convex optimization with long-term constraints, which extends Yuan & Lamperski (2018)’s work to decentralized case with time-varying directed network. The authors propose DOCO frameworks (full-information and one-point bandit feedback) based on augmented Lagrangian functions. They also provide the corresponding regret bounds for both strongly and non-strongly convex cases. The experiments on synthetic data validate the effectiveness of proposed algorithms.

The problem setting of this paper is interesting and the theoretical contribution is nice, but the empirical studies could be improved:

1. It is prefer to append some experiments on real-world applications.

2. Although the regret bound of DOCO is better, the projection step is expensive. Can you compare the running time of DOCO with projection-free algorithms?


**Experience Assessment:**

I have read many papers in this area.

**Review Assessment: Checking Correctness Of Derivations And Theory:**

I assessed the sensibility of the derivations and theory.

**Review Assessment: Checking Correctness Of Experiments:**

I carefully checked the experiments.

**Review Assessment: Thoroughness In Paper Reading:**

I read the paper at least twice and used my best judgement in assessing the paper.

---

> ### Author Response · Authors · 2019-11-08
> **Response to Official Blind Review #2**
>
> We thank the Reviewer for the positive comments and the suggestions.
>
> 1. It is prefer to append some experiments on real-world applications.
>
> We have followed your suggestion and appended some experiments on real-world datasets, which are selected from the LIBSVM repository ( https://www.csie.ntu.edu.tw/~cjlin/libsvmtools/datasets/ ). More specifically, we investigated the performance of our algorithm on the mg dataset that has 6 features and 1385 instances, and on the bodyfat dataset that has 14 features and 252 instances. The plots are provided in the following screenshots (mg dataset with \rho = 0, please see https://pasteboard.co/IFy9B7U.png ; mg dataset with \rho = 1, please see https://pasteboard.co/IFyajAf.png ; bodyfat dataset with \rho = 0, please see, https://pasteboard.co/IFyaIPX.png ; bodyfat dataset with \rho = 1, please see https://pasteboard.co/IFyb9TY.png ). We have used the same network as that used in Section 4 of the paper.
>
> These numerical experiments on real-world datasets show the convergence of the proposed algorithms and are consistent with the results established in Theorems 2 and 4 of our paper.
>
> 2. Although the regret bound of DOCO is better, the projection step is expensive. Can you compare the running time of DOCO with projection-free algorithms?
>
> Observe that the projection step in our algorithms is really inexpensive, as it consists in projecting onto a ball. Specifically, to avoid projections is to allow the algorithm to violate the constraints by projecting onto a simplified set (a ball) which covers the original constraint set. In every round, each node projects its decision onto a Euclidean ball, which can be easily solved analytically, but at the price of constraint violations.
>
> We thank the Reviewer for pointing out the projection-free algorithms to us, which is indeed quite relevant. In fact, we cited a most related paper and made comparisons with it in the manuscript. We have made some comparisons with the projection-free algorithm (i.e., D-OCG in Wenpeng Zhang, et al., ICML, 2017) using two real-world datasets selected from the LIBSVM repository (https://www.csie.ntu.edu.tw/~cjlin/libsvmtools/datasets/ ). The details of the datasets are summarized as follow:
> ----------------------------------------------------------------
> Dataset  |    # of features    |    # of instances
> ----------------------------------------------------------------
> mg          |             6               |       1385
> ----------------------------------------------------------------
> bodyfat  |            14              |        252
> ----------------------------------------------------------------
>
> We use the same network as that used in Section 4 of the paper. The detailed results are provided in the following screenshots (mg dataset with \rho = 1, please see https://pasteboard.co/IFBMgLM.png ; bodyfat dataset with \rho = 1, please see https://pasteboard.co/IFBMHDE.png ).
>
> From these plots one can confirm that: (i) Our DOCO algorithm achieves better performance than D-OCG under the same information feedback (of course, D-OCG exhibits no constraint violations); and (ii) For both algorithms, the performance is degraded from full information to bandit feedback.

---

> > ### Comment · AnonReviewer2 · 2019-11-12
> > **I have read the rebuttal**
> >
> > Thanks for your response. I believe the additional empirical studies make the paper more convincing.

---

### Official Review · AnonReviewer3 · 2019-10-24
**Official Blind Review #3**

**Rating:** 3

**Review:**

The authors study distributed online convex optimization where the distributed system consists of various computing units connected by a time varying graph. The authors prove optimal regret bounds for a proposed decentralized algorithm and experimentally evaluate the performance of their algorithms on distributed online regularized linear regression problems.

The paper seems well written and well researched and places itself well in context of current literature. The authors also improve the state of the art in the field. The main weakness of the paper is the limited experimental evaluation and applicability of the assumptions and the theoretical setting that underpins this work.

[Edit: After going through the other reviews, I have downgraded my score. The revised version of the paper the authors uploaded is 23 pages long with the main paper body being 10 pages. The CFP instructs reviewers to apply a higher standard to judge such long papers. I  am not convinced that the paper is solving an important problem that merits such a long paper.]

**Experience Assessment:**

I do not know much about this area.

**Review Assessment: Checking Correctness Of Derivations And Theory:**

I did not assess the derivations or theory.

**Review Assessment: Checking Correctness Of Experiments:**

I assessed the sensibility of the experiments.

**Review Assessment: Thoroughness In Paper Reading:**

I made a quick assessment of this paper.

---

> ### Author Response · Authors · 2019-11-08
> **Response to Official Blind Review #3**
>
> We thank the Reviewer for the positive comments.
>
> About our assumptions. Assumption 4 is quite standard in the literature on distributed online or offline optimization, and easy to achieve in a distributed manner in real-world networks. For example, when bidirectional communication between nodes is allowed, we can enforce symmetry on the node interaction matrix, which immediately makes it doubly stochastic. There are also other methods to construct doubly stochastic matrices for a network, see, e.g., [F. Garin and L. Schenato. A survey on distributed estimation and control applications using linear consensus algorithms, 2011] and [Bahman Gharesifard and Jorge Cortes. When does a digraph admit a doubly stochastic adjacency matrix?, 2010].
>
> About real-world experiments. We have followed the Reviewer’s suggestion and implemented our algorithms over the distributed online regularized linear regression problem with two real datasets selected from the LIBSVM repository ( https://www.csie.ntu.edu.tw/~cjlin/libsvmtools/datasets/ ). The details of the datasets are summarized as follows:
> ----------------------------------------------------------------
> Dataset  |    # of features    |    # of instances
> ----------------------------------------------------------------
> mg          |             6               |       1385
> ----------------------------------------------------------------
> bodyfat  |            14              |        252
> ----------------------------------------------------------------
>
> We use the same network as that used in Section 4 of the paper. The plots are provided in the following screenshots (mg dataset with \rho = 0, please see https://pasteboard.co/IFy9B7U.png ; mg dataset with \rho = 1, please see https://pasteboard.co/IFyajAf.png ; bodyfat dataset with \rho = 0, please see, https://pasteboard.co/IFyaIPX.png ; bodyfat dataset with \rho = 1, please see https://pasteboard.co/IFyb9TY.png ). These numerical experiments on real-world datasets show the convergence of the proposed algorithms and are consistent with the results established in Theorems 1-4 of our paper.
>
> Furthermore, to demonstrate the efficiency of our algorithm, we have compared our algorithm with a standard distributed online optimization algorithm called D-OCG (Distributed Online Conditional Gradient) on these two real datasets. The detailed results are provided in the screenshots (mg dataset with \rho = 1, please see https://pasteboard.co/IFBMgLM.png ; bodyfat dataset with \rho = 1, please see https://pasteboard.co/IFBMHDE.png ) .

---

### Official Review · AnonReviewer1 · 2019-10-26
**Official Blind Review #1**

**Rating:** 6

**Review:**

Summary:
The paper considers a distributed variant of online convex optimization problem over multiple players, where, at each trial t, convex loss_l_t.i is revealed to player i and but evaluated by sum of loss functions sum_i=1^n l_t,i. The players can communicate with their neighborhood and share their decisions. Under the problems setting and some assumption on the neighborhood graph structure, the authors prove regret bounds for convex/strongly-convex losses and full-info/bandit settings. Specifically, the paper allows the algorithm to violate domain constraints but the sum of violation has to be sublinear in the number of trials. They also show the violation bounds simultaneously.


Comments:
The key assumption is Assumption 4, that the players share a doubly-stochastic matrix which is used to mix neighbors’ decisions. This assumption allows to mix all players’ decisions in a long run and the derived regret bounds make sense. The theoretical results are non-trivial.

As a summary, I feel the results are beyond standard previous work and has certain values. Note that I have not evaluated correctness of the results, but the results are likely under Assumption 4.


**Experience Assessment:**

I have published in this field for several years.

**Review Assessment: Checking Correctness Of Derivations And Theory:**

I assessed the sensibility of the derivations and theory.

**Review Assessment: Checking Correctness Of Experiments:**

I did not assess the experiments.

**Review Assessment: Thoroughness In Paper Reading:**

I read the paper at least twice and used my best judgement in assessing the paper.

---

> ### Author Response · Authors · 2019-11-08
> **Response to Official Blind Review #1**
>
> We thank the Reviewer for these insightful comments. We would like to emphasize that Assumption 4 is quite standard in the literature on distributed online or offline optimization, and easy to achieve in a distributed manner in real-world networks. For example, when bidirectional communication between nodes is allowed, we can enforce symmetry on the node interaction matrix, which immediately makes it doubly stochastic. There are also other methods to construct doubly stochastic matrices for a network, see, e.g., [F. Garin and L. Schenato. A survey on distributed estimation and control applications using linear consensus algorithms, 2011] and [Bahman Gharesifard and Jorge Cortes. When does a digraph admit a doubly stochastic adjacency matrix?, 2010].

---

### Author Response · Authors · 2019-11-08
**Revision uploaded**

We thank all the reviewers for their invaluable and constructive feedback. We have replied to each reviewer's concerns separately, and uploaded a revised version of our paper.

---

### Decision · Program_Chairs · 2019-12-19

**Decision:**

Reject

**Comment:**

The paper proposes  a decentralized algorithm with regret for distributed online convex optimization problems. The reviewers worry about the assumptions and the theoretical settings, they also find that the experimental evaluation  is insufficient.